# Optimal Brain Apoptosis

**Mingyuan Sun**[1], **Zheng Fang**[1,*], **Jiaxu Wang**[2], **Junjie Jiang**[1], **Delei Kong**[3],
**Chenming Hu**[1], **Yuetong Fang**[2] **& Renjing Xu**[2,*]
[1]Northeastern University     [2]The Hong Kong University of Science and Technology (Guangzhou)
[3]Hunan University

## Abstract

The increasing complexity and parameter count of Convolutional Neural Networks (CNNs) and Transformers pose challenges in terms of computational efficiency and resource demands. Pruning has been identified as an effective strategy to address these challenges by removing redundant elements such as neurons, channels, or connections, thereby enhancing computational efficiency without heavily compromising performance. This paper builds on the foundational work of Optimal Brain Damage (OBD) by advancing the methodology of parameter importance estimation using the Hessian matrix. Unlike previous approaches that rely on approximations, we introduce Optimal Brain Apoptosis (OBA), a novel pruning method that calculates the Hessian-vector product value directly for each parameter. By decomposing the Hessian matrix across network layers and identifying conditions under which inter-layer Hessian submatrices are non-zero, we propose a highly efficient technique for computing the second-order Taylor expansion of parameters. This approach allows for a more precise pruning process, particularly in the context of CNNs and Transformers, as validated in our experiments including VGG19, ResNet32, ResNet50, and ViT-B/16 on CIFAR10, CIFAR100 and Imagenet datasets. Our code is available at https://github.com/NEU-REAL/OBA.

## 1 Introduction

With the rapid development of deep learning, neural networks have become deeply integrated into all sectors of our daily life. Convolutional Neural Networks (LeCun et al., 1998; Krizhevsky et al., 2012; He et al., 2016; Ma et al., 2023) and Transformers (Vaswani et al., 2017; Dosovitskiy et al., 2021) are two typical structures used most widely. As researchers continuously innovate, the performance of neural networks improves, but the number of parameters and the computational complexity also increase significantly. Therefore, how to efficiently reduce the parameter size and computational overhead of neural networks while maintaining their performance as much as possible has become a crucial problem.

Extensive research (LeCun et al., 1989; Molchanov et al., 2016) demonstrates that pruning is a powerful tool for dealing with this issue. Generally speaking, pruning can be divided into two main streams: unstructured pruning and structured pruning. Unstructured pruning (Guo et al., 2016; Han et al., 2015; Dong et al., 2017) involves the removal of individual weights or neurons from a neural network. The key advantage of unstructured pruning lies in its flexibility and the fine-grained control it offers over the model's architecture. This method often requires specialized hardware or software to exploit the resultant sparsity for computational efficiency. Structured pruning (Anwar et al., 2017; Yeom et al., 2021) removes entire neurons, channels, or layers from a neural network, which is more frendly to software since parameters of neural networks are mainly structured data, such as tensor, matrix, and vector. Removing entire neurons directly corresponds to a slicing or selecting operation on the structured data, which is easy to implement and more compatible with standard hardware accelerators, such as GPUs and TPUs.

Hanson & Pratt (1988) is one of the earliest works to explore structured pruning. The underlying idea is that significant weights typically possess greater magnitudes, as they need to process and transmit

---

*Corresponding authors.

more information to be influential in determining the network's accurate output. However, pruning under this guidance may sometimes incorrectly remove important neurons with small magnitude and reserve unimportant neurons with large magnitude. Optimal Brain Damage (OBD) (LeCun et al., 1989) and Optimal Brain Surgeon (OBS) (Hassibi & Stork, 1992) propose to leverage the Hessian matrix of loss w.r.t. parameters to estimate the importance of each parameter. The Hessian matrix contains second-order partial derivatives of the loss function w.r.t. all pairs of parameters, which is very computationally expensive to compute. Thus, OBD approximates it as a diagonal matrix by assuming the loss of deleting several parameters is the sum of deleting these parameters individually. OBS views the pruning as an optimization problem and solves it with the Lagrange multiplier. They either discard or approximate the second-order partial derivatives between all pairs of parameters, which capture the change of loss on one parameter when deleting another parameter.

**Our Contributions** In this paper, we follow the idea of OBD, leveraging Hessian matrix for parameter importance estimation. Instead of approximating Hessian matrix, we calculate the Hessian-vector product element $\sum_j \frac{\partial^2 \mathcal{L}}{\partial \theta_i \partial \theta_j} \delta \theta_i \delta \theta_j$ for each parameter in the network. To achieve this, we first separate the Hessian matrix of the whole network into Hessian submatrices between layers. Then, in the context of widely used network structures including convolutional neural networks (CNNs) and Transformers, we analyze the conditions where the Hessian submatrices between two layers are nonzero. Finally, we propose a highly efficient method to capture these conditions and obtain the Hessian-vector product element on each parameter. Stepping from approximating the Hessian matrix with the Fisher matrix to directly computing the Hessian-vector product, we propose Optimal Brain Apoptosis (OBA), a novel pruning method that efficiently calculates the second-order Taylor expansion for each parameter and is applicable to both structured and unstructured pruning tasks.

## 2 RELATED WORK

**Model Compression** Model compression is an area that focuses on creating smaller, faster, and more efficient models suitable for deployment in environments with limited resources, like mobile devices or embedded systems. There are several typical fields within this area, including quantization (Courbariaux et al., 2015; Rastegari et al., 2016; Pouransari et al., 2020), knowledge distillation (Hinton et al., 2015; Chen et al., 2021; Zhou et al., 2021), neural architecture search (Liu et al., 2018; Zoph & Le, 2016; Pham et al., 2018), and network pruning (Molchanov et al., 2019; 2016). Quantization, outlined in works like Hubara et al. (2018) and Jacob et al. (2018), focuses on reducing parameter precision to accelerate inference and decrease model size, enabling deployment on devices with limited resources. Knowledge distillation, as introduced by Hinton et al. (2015); Romero et al. (2014), leverages a smaller "student" model to mimic a larger "teacher" model, effectively compressing the knowledge and achieving high performance with less computational demand. Neural Architecture Search (NAS), with seminal contributions from Zoph & Le (2016), automates the discovery of optimal architectures, often outperforming human-designed models in efficiency and accuracy. Pruning techniques, highlighted in work by Han et al. (2015), remove non-essential weights or neurons, significantly reducing model complexity and enhancing inference speed without major accuracy losses. Together, these techniques represent the forefront of model compression research, addressing the balance between performance and computational efficiency necessary for advanced AI applications.

**Network Pruning** Network pruning, initially recognized as an importance estimation problem (Molchanov et al., 2019; Chauvin, 1988; Yu et al., 2018; He et al., 2020), has been prompting researchers to focus on finding accurate criteria that reveals the importance of parameters or neurons in neural networks. Molchanov et al. (2016) operated under the assumption that each layer in feed-forward networks held equal importance and introduced a heuristic for global scaling normalization. However, this approach did not prove effective in networks incorporating skip connections. Additionally, the method relies on using network activations to calculate its criterion, resulting in increased memory demands. In contrast, pruning methods that focus on batch normalization (Gordon et al., 2018; Huang & Wang, 2018; Liu et al., 2017; Ye et al., 2018) bypass the need for sensitivity analysis and are applicable on a global scale. In intricate network architectures (Liu et al., 2021; Luo & Wu, 2020; You et al., 2019; Zhang et al., 2021), parameter interdependencies often require their joint pruning. This collective pruning of interlinked parameters has been an area of focus in structured pruning research since its early stages. Fang et al. (2023) proposes to build a dependency

graph which captures the interdependencies between parameters and prunes parameters belonging to a graph together to achieve structured pruning for neural networks with complicated structures. In our pruning setting, we view the pruning as an importance estimation problem for each individual parameter (unstructured) or parameter group (structured) obtained from Fang et al. (2023). Parameters with low importance are pruned in each pruning step.

**Hessian Matrix Estimation** The structure and computational aspects of the Hessian matrix in feedforward neural networks have been extensively studied since the early 1990s (Buntine & Weigend, 1994; Wille, 1997). The Hessian matrix was first utilized in neural network pruning by LeCun et al. (1989) to calculate the importance score of each neuron, leveraging diagonal approximation is used to estimate the Hessian matrix:

$$\delta\mathcal{L}_{\text{OBD}} = \frac{1}{2}\left(\boldsymbol{\theta}_q^*\right)^2 \mathbf{H}_{qq}. \tag{1}$$

Building upon this, OBS (Hassibi & Stork, 1992) views the importance estimation as an optimization problem, and aims at finding a set of weights that yields least change on the loss:

$$\min_q \left\{ \min_{\delta\boldsymbol{\theta}} \frac{1}{2}\delta\boldsymbol{\theta}^\top \mathbf{H}\delta\boldsymbol{\theta} \text{ s.t. } \mathbf{e}_q^\top \delta\boldsymbol{\theta} + \boldsymbol{\theta}_q^* = 0 \right\}. \tag{2}$$

Early research such as that by Buntine & Weigend (1994) provided an extensive review of how to compute second derivatives in feed-forward networks, and Wille (1997) examined the Hessian matrix's structure and second derivative techniques. More contemporary efforts, such as EigenDamage (Wang et al., 2019), utilize a Kronecker-factored eigenbasis for network reparameterization and approximate the Hessian matrix using the Fisher matrix, as discussed by Martens (2020). Recent studies (Wu et al., 2020; Singh et al., 2021) have thoroughly investigated the common structures and rank properties of the Hessian in neural networks. Pearlmutter (1994) initially introduced an efficient method for computing the Hessian-vector product in feedforward neural networks. Our research applies this idea to the pruning of modern network architectures including CNNs and Transformers.

## 3 PRELIMINARY

Consider a feed-forward neural network with parameters $\theta$ and $L$ layers. Similar to OBD (LeCun et al., 1989), when we add small perturbation $\delta\theta$ on $\theta$, the second-order Taylor expansion of the perturbation on the objective function is given by

$$\delta\mathcal{L}(\theta) = (\frac{\partial\mathcal{L}}{\partial\theta})^\mathsf{T}\delta\theta + \frac{1}{2}\delta\theta^\mathsf{T}\mathbf{H}\delta\theta + o(\|\delta\theta\|^3), \tag{3}$$

where $\mathbf{H}$ is the Hessian matrix that represents the second-order derivatives between all parameter pairs. It is usually infeasible to compute the Hessian matrix due to its complexity of $O(n^2)$, where n is the number of parameters in the network (LeCun et al., 1989; Hassibi & Stork, 1992). By expanding the first and second term of eq. (3), we can define the perturbation of the loss caused by $\delta\theta_i$ as

$$\delta\mathcal{L}(\theta_i) = \frac{\partial\mathcal{L}}{\partial\theta_i}\delta\theta_i + \sum_j \frac{\partial^2\mathcal{L}}{\partial\theta_i\partial\theta_j}\delta\theta_i\delta\theta_j + o(\|\delta\theta_i\|^3). \tag{4}$$

The first term of eq. (4) is leveraged to estimate the improtance of neurons in Molchanov et al. (2016). Same to prior works, we ignore the higher order term $o(\|\delta\theta_i\|^3)$. Current works (Hassibi & Stork, 1992; Yu et al., 2022; Benbaki et al., 2023) that leverage the second-order Taylor expansion term approximate the Hessian matrix with Fisher information matrix. This approximation, if applied to eq. (4), would change its second-order term to $\sum_j \frac{\partial\mathcal{L}}{\partial\theta_i}\frac{\partial\mathcal{L}}{\partial\theta_j}\delta\theta_i\delta\theta_j$. However, this approximation is not accurate enough to capture the second-order loss perturbation caused by $\delta\theta_i$ and $\delta\theta_j$. Therefore, we focus on a theoretical analysis on how to calculate the original second-order term $\sum_j \frac{\partial^2\mathcal{L}}{\partial\theta_i\partial\theta_j}\delta\theta_i\delta\theta_j$.

## 4 METHOD

### 4.1 DEFINITION

We derive from a general form of linear layers in modern neural networks. For layer $l \in [1, L]$, we denote the weight parameter as $W^{(l)} \in \mathbb{R}^{l_{\text{out}} \times l_{\text{in}} \times p_{\text{weight}}}$ and bias parameter as $b^{(l)} \in \mathbb{R}^{l_{\text{out}}}$. We denote

the input of layer $l$ as $X^{(l)} \in \mathbb{R}^{l_{in} \times p_{in}}$ and output of layer $l$ as $Y^{(l)} \in \mathbb{R}^{l_{out} \times p_{out}}$. $p_{weight}$, $p_{out}$, and $p_{in}$ are the length of flattened weights, output values, and input values that contribute to the connections between every pairs of input neurons and output neurons, for example $p_{weight} = 1, p_{out} = 1, p_{in} = 1$ in the context of fully connected layers and $p_{weight} = s_{kernel}^{(l)\, 2}, p_{out} = h_{out}^{(l)} w_{out}^{(l)}, p_{in} = h_{in}^{(l)} w_{in}^{(l)}$ in the context of convolutional layers. The forward propagation of layer $l$ is given by

$$Y_{ab}^{(l)} = \sum_{cde} W_{acd}^{(l)} X_{ce}^{(l)} M_{bde}^{(l)} + b_a^{(l)}, \tag{5}$$

where $M \in \{0,1\}^{p_{out} \times p_{weight} \times p_{in}}$, determined by the layer itself, represents the connections among all input values, weight values, and output values. We denote the flattened vector of $X^{(l)}$, $W^{(l)}$, $Y^{(l)}$, and $M^{(l)}$ as $x^{(l)} \in \mathbb{R}^{l_{in} \cdot p_{in}}$, $w^{(l)} \in \mathbb{R}^{l_{out} \cdot l_{in} \cdot p_{weight}}$, $y^{(l)} \in \mathbb{R}^{l_{out} \cdot p_{out}}$, and $m^{(l)} \in \{0,1\}^{p_{out} \cdot p_{weight} \cdot p_{in}}$, respectively. the parameters $\theta^{(l)}$ of layer $l$ can be expressed as $\theta^{(l)} = \begin{bmatrix} w^{(l)} \\ b^{(l)} \end{bmatrix}$. We represent all partial derivative terms with variables in their vector forms to ensure these terms are with the same definition of Jacobian matrix.

In a certain layer $l$ defined by eq. (5), the gradient of the loss w.r.t. the indexed parameters $W_{acd}^{(l)}$ and $b_a^{(l)}$ are respectively $\sum_{ab} \frac{\partial \mathcal{L}}{\partial Y_{ab}^{(l)}} \sum_e X_{ce}^{(l)} M_{bde}^{(l)}$ and $\sum_b \frac{\partial \mathcal{L}}{\partial Y_{ab}^{(l)}}$. It is clear that the condition for the Hessian matrix entries between layer $l$ and any distinct layer $l'$ being exclusively zero hinges on the differentiability of terms $\frac{\partial \mathcal{L}}{\partial y^{(l)}}$ and $X^{(l)}$ with respect to the parameters of layer $l'$, which depends on the connection type of two layer types. We observe that the connectivity types which introduce nonzero Hessian submatrices between any two layers in a neural network can be divided into two cases: series connectivity and parallel connectivity, as shown in fig. 1a. We next analyze these two cases and calculate the Hessian-vector product element $\sum_j \frac{\partial^2 \mathcal{L}}{\partial \theta_i \partial \theta_j} \delta\theta_i \delta\theta_j$ in eq. (4) for the two cases respectively.

## 4.2 SERIES CONNECTIVITY

**Definition 4.1** (Series Connectivity). In a neural network at layer $l$, if there exists a layer $l'$ such that there is a differentiable function mapping the output of layer $l'$ to the input of layer $l$, we say layer $l'$ and $l$ are in series connectivity. Specifically:

- layer $l'$ is in lower series connectivity to layer $l$.
- layer $l$ is in upper series connectivity to layer $l'$.

In fig. 1a, there are differentiable functions from $Y^{(l_1)}$ to $X^{(l_2)}$ and $X^{(l_3)}$, respecively, so layer $l_1$ is in series connectivity with both layer $l_2$ and layer $l_3$. Without loss of generality, we take layer $l_1$ and layer $l_2$ as an example. Then

$$\frac{\partial \mathcal{L}}{\partial y^{(l_1)}} = \frac{\partial \mathcal{L}}{\partial y^{(l_2)}} \underbrace{\frac{\partial y^{(l_2)}}{\partial x^{(l_2)}}}_{\text{differentiable to } \theta^{(l_2)}} \frac{\partial x^{(l_2)}}{\partial y^{(l_1)}},$$

also $X^{(l_2)}$ is differentiable to $\theta^{(l_1)}$. According to our analysis in the beginning of this section, a nonzero Hessian submatrix exists between layers $l_1$ and $l_2$.

**Theorem 4.2.** *For layer $l$ in a neural network where layers $l_{up} \in \mathbf{L}_{up}$ and layers $l_{low} \in \mathbf{L}_{low}$ are in upper and lower series connectivity to layer $l$, respectively, then for weight parameter $w^{(l)}$ and bias parameter $b^{(l)}$ of layer $l$, we have*

$$\sum_{l' \in \mathbf{L}_{up} \cup \mathbf{L}_{low}} \sum_j \frac{\partial^2 \mathcal{L}}{\partial \theta_j^{(l')} \partial w^{(l)}} \delta\theta_j^{(l')} \odot \delta w^{(l)} = \sum_{l_{up} \in \mathbf{L}_{up}} \frac{\partial \mathcal{L}}{\partial y^{(l_{up})}} \mathbf{J}_{\delta W^{(l_{up})}}^{(l_{up})} \frac{\partial x^{(l_{up})}}{\partial w^{(l)}} \odot \delta w^{(l)} \tag{6}$$

$$+ \frac{\partial \mathcal{L}}{\partial y^{(l)}} \frac{\partial y^{(l)}}{\partial w^{(l)}} \bigg|_{\hat{X}^{(l)}} \odot \delta w^{(l)} \tag{7}$$

*in which $\hat{X}^{(l)}$ is given by*

$$\hat{X}_{mn}^{(l)} = \sum_{l_{low} \in \mathbf{L}_{low}} \sum_k \frac{\partial X_{mn}^{(l)}}{\partial \theta_k^{(l_{low})}} \delta\theta_k^{(l_{low})}, \tag{8}$$

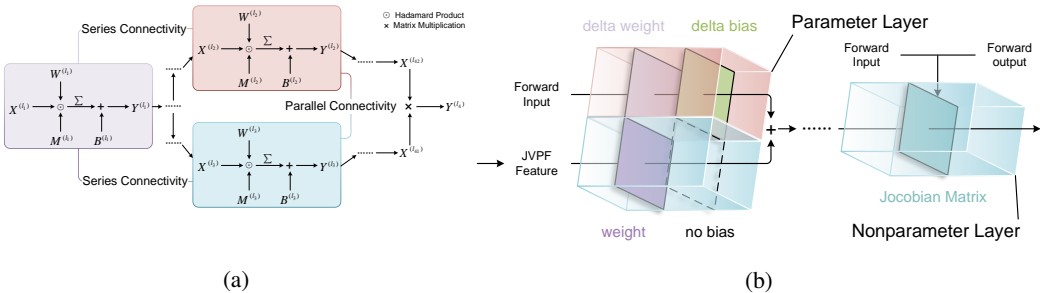

| (a) | (b) |
|---|---|

Figure 1: **(a)** An illustration of conditions where the Hessian matrix between parameters of two layers are nonzero. **(b)** An illustration of Jacobian-Vector Product Forward Propagation. Two forward propagation processes are needed for parameter layers and one forward propagation process is needed for nonparameter layers. For nonparameter layers we leverage Jacobian-vector product to conduct the forward process and do not need to calculate the Jacobian matrix explicitly.

*and*

$$\sum_{l' \in \mathbf{L}_{up} \cup \mathbf{L}_{low}} \sum_{j} \frac{\partial^2 \mathcal{L}}{\partial \theta_j^{(l')} \partial b^{(l)}} \delta\theta_j^{(l')} \odot \delta b^{(l)} = \sum_{l_{up} \in \mathbf{L}_{up}} \frac{\partial \mathcal{L}}{\partial y^{(l_{up})}} \mathbf{J}_{\delta W^{(l_{up})}}^{(l_{up})} \frac{\partial x^{(l_{up})}}{\partial b^{(l)}} \odot \delta b^{(l)}, \tag{9}$$

*where* $\mathbf{J}_{\delta W^{(l)}}^{(l)} \in \mathbb{R}^{(l_{out} \cdot m_{out}) \times (l_{in} \cdot m_{in})}$ *is the jacobian matrix of* $y^{(l)}$ *with respect to* $x^{(l)}$ *taking* $\delta W^{(l)}$ *as the weights, and* $\frac{\partial y^{(l)}}{\partial w^{(l)}}\big|_{\hat{X}^{(l)}}$ *is the jacobian matrix of* $y^{(l)}$ *w.r.t.* $w^{(l)}$ *taking* $\hat{X}^{(l)}$ *as input.*

The proof is provided in appendix E.1. Note that the sets $\mathbf{L}_{up}(l)$ and $\mathbf{L}_{low}(l)$ are dependent on layer $l$, and are abbreviated for simplicity. With theorem 4.2, the second-order term of the Taylor expansion of the loss with respect to each individual parameter can be computed, taking into account parameters belonging to layers that are in series connectivity to a specific layer. For classical neural network structures such as convolutional neural networks and fully connected neural networks, there only exist series connectivities between layers. Thus, we can directly apply theorem 4.2 to calculate eq. (4) for each individual parameter.

## 4.3 PARALLEL CONNECTIVITY

For recent novel neural network structures such as Transformer (Vaswani et al., 2017), matrix multiplication plays a crucial and effective role in achieving their impressive performance. It also introduces parallel connectivity to layers and lead to the nonzero Hessian matrices between these connected layers.

**Definition 4.3.** In a neural network, if there exist two layers $l$ and $l'$ such that there are differentiable functions respectively mapping the outputs of layer $l$ and $l'$ to the inputs $X^{(left)}$ and $X^{(right)}$ of a matrix multiplication operation $Y^{(mul)} = X^{(left)} X^{(right)}$, i.e., layers $l$ and $l'$ are both in lower series connectivity to the matrix multiplication operation, we say layer $l$ and $l'$ are in parallel connectivity. Denote the multiplication operation as layer $l_m$, then

- layer $l$ is in left parallel connectivity to layer $l'$.
- layer $l'$ is in right parallel connectivity to layer $l$.

In fig. 1a, layer $l_2$ and layer $l_3$ are in parallel connectivity. Similarly, the gradient of loss w.r.t. parameters of layer $l_2$ is $\frac{\partial \mathcal{L}}{\partial \theta^{(l_2)}} = \frac{\partial \mathcal{L}}{\partial y^{(l_4)}} \frac{\partial y^{(l_4)}}{\partial y^{(l_{42})}} \frac{\partial y^{(l_{42})}}{\partial \theta^{(l_2)}}$ in which $\frac{\partial y^{(l_4)}}{\partial y^{(l_{42})}}$ is differentiable to $\theta^{(l_3)}$ and vice versa. Therefore, the nonzero Hessian submatrices resulting from parallel connectivity should also be considered.

**Theorem 4.4.** *For a multiplication operation* $Y^{(mul)} = X^{(left)} X^{(right)}$ *in a neural network, where* $X^{(left)} \in \mathbb{R}^{l_{row} \times l_{hid}}$ *and* $X^{(right)} \in \mathbb{R}^{l_{hid} \times l_{col}}$, *layers* $l_l \in \mathbf{L}_{left}$ *and* $l_r \in \mathbf{L}_{right}$ *are in lower series connectivity to this multiplication operation, respectively. Consider two surrogate weight matrices*

$\hat{X}^{(left)} \in \mathbb{R}^{l_{row} \times l_{hid}}$ and $\hat{X}^{(right)} \in \mathbb{R}^{l_{hid} \times l_{col}}$ given by:

$$\hat{X}_{kn}^{(left)} = \sum_{l_l \in \mathbf{L}_{left}} \sum_q \frac{\partial X_{kn}^{(left)}}{\partial \theta_q^{(l_l)}} \delta\theta_q^{(l_l)}, \tag{10}$$

$$\hat{X}_{no}^{(right)} = \sum_{l_r \in \mathbf{L}_{right}} \sum_r \frac{\partial X_{no}^{(right)}}{\partial \theta_r^{(l_r)}} \delta\theta_r^{(l_r)}. \tag{11}$$

*Then we have*

$$\sum_{l_l \in \mathbf{L}_{left}} \sum_q \frac{\partial^2 \mathcal{L}}{\partial \theta_q^{(l_l)} \partial \theta^{(l_r)}} \delta\theta_q^{(l_l)} \odot \delta\theta^{(l_r)} = \frac{\partial \mathcal{L}}{\partial y^{(mul)}} \frac{\partial y^{(mul)}}{\partial x^{(right)}} \bigg|_{\hat{X}^{(left)}} \frac{\partial x^{(right)}}{\partial \theta^{(l_r)}} \odot \delta\theta^{(l_r)} \tag{12}$$

*and*

$$\sum_{l_r \in \mathbf{L}_{right}} \sum_r \frac{\partial^2 \mathcal{L}}{\partial \theta_r^{(l_r)} \partial \theta^{(l_l)}} \delta\theta_r^{(l_r)} \odot \delta\theta^{(l_l)} = \frac{\partial \mathcal{L}}{\partial y^{(mul)}} \frac{\partial y^{(mul)}}{\partial x^{(left)}} \bigg|_{\hat{X}^{(right)}} \frac{\partial x^{(left)}}{\partial \theta^{(l_l)}} \odot \delta\theta^{(l_l)}. \tag{13}$$

Proof can be seen in appendix E.2. Now we can analytically calculate the Hessian-vector product element of each parameter specifically focusing on the interaction between its layer and any other layers that are in parallel connectivity to it. Combining theorem 4.2 and theorem 4.4, we can calculate the Hessian-vector product element of each parameter in a neural network. Next, we focus on an efficient calculation the Hessian-vector product element of each parameter.

## 4.4 JACOBIAN-VECTOR PRODUCT FORWARD PROPAGATION

From a computational overhead perspective, the main calculation part within theorem 4.2 and theorem 4.4 focuses on $\hat{X}^{(l)}$, the gradients of $X^{(l)}$ w.r.t. the parameters of all layers that are in lower series connectivity to $\hat{X}^{(l)}$, as we need to separately back-propagate each entry of $X^{(l)}$, usually a large matrix in attention modules. To address this issue, we introduce Jacobian-Vector Product Forward Propagation (JVPF), a method capable of computing $\hat{X}^{(l)}$ for all layers with an acceptable computational expense.

Let's look at a layer in lower series connectivity to layer $l$ and denote it as layer $l_{\text{low}}$. The derivative of $X^{(l)}$ w.r.t. the parameter $\theta^{(l_{\text{low}})}$ a single layer $l_{\text{low}}$ multiplied with $\delta\theta^{(l_{\text{low}})}$ can be expressed as

$$\frac{\partial x^{(l)}}{\partial \theta^{(l_{\text{low}})}} \delta\theta^{(l_{\text{low}})} = \frac{\partial x^{(l)}}{\partial x^{(l-1)}} \cdots \frac{\partial x^{(l_{\text{low}}+1)}}{\partial x^{(l_{\text{low}})}} \frac{\partial x^{(l_{\text{low}})}}{\partial \theta^{(l_{\text{low}})}} \delta\theta^{(l_{\text{low}})},$$

indicating that we can calculate it in a layer-by-layer manner. Further, let us group layers in lower series connectivity to layer $l$ into several groups, each of which contains both parameter layers and nonparameter layers that are in series connectivity to each other. We denote the $i^{th}$ group as $\mathbf{G}_i = \{l_{i1}, l_{i2}, \cdots, l_{iN_i}\}$, where $N_i$ is the number of layers in group $\mathbf{G}_i$ and layer $l_{i(j+1)}$ is subsequent to layer $l_{ij}$. Then we have

$$\hat{x}^{(l)} = \sum_i \sum_{l_{\text{low}} \in \mathbf{G}_i} \frac{\partial x^{(l)}}{\partial \theta^{(l_{\text{low}})}} \delta\theta^{(l_{\text{low}})} = \sum_i \frac{\partial x^{(l)}}{\partial y^{(l_{N_i})}} f^{(l_{iN_i})} \circ f^{(l_{i(N_i-1)})} \circ \cdots \circ f^{(l_{i1})}(0)$$

where

$$f^{(l_{ij})}(x) = \begin{cases} \frac{\partial y^{(l_{ij})}}{\partial \theta^{(l_{ij})}} \delta\theta^{(l_{ij})} + \frac{\partial y^{(l_{ij})}}{\partial x^{(l_{ij})}} x & l_{ij} \text{ has parameters,} \\ \frac{\partial y^{(l_{ij})}}{\partial x^{(l_{ij})}} x & \text{else.} \end{cases} \tag{14}$$

$x^{(l_{ij})}$, $y^{(l_{ij})}$, and $\theta^{(l_{ij})}$ are the input, output, and parameters of layer $l_{ij}$. Once we replace the forward function of each layer with eq. (14), we could calculate $\hat{x}^{(l)}$ for each layer $l$ through one forward propagation process, and all series connectivity groups can be calculated in parallel. An intuitive demonstration of JVPF is shown in fig. 1b.

## 4.5 PRUNING STRATEGY

Utilizing theorem 4.2 and theorem 4.4 along as our proposed JVPF, which offers an efficient choice to obtain the essential intermediate values, we can calculate the Hessian-vector product element of each parameter as their importance scores with several batches of training data. Please refer to appendix B for the detailed importance score acquisition. This importance score can be leveraged to conduct unstructured pruning for each parameter individually, or conduct structured pruning for each parameter group.

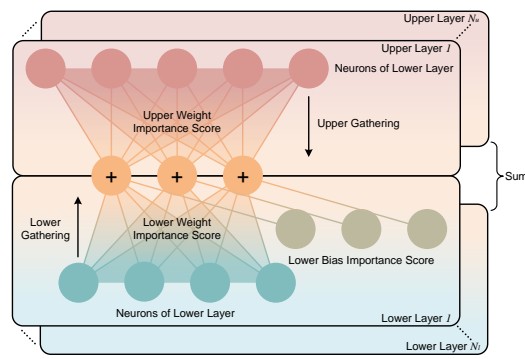

Figure 2: Importance score of each neuron in a group is gathered from parameters of lower layers and upper layers.

**Structured Pruning** Following Fang et al. (2023), parameters from in-layer and interlayer connections can be organized into several groups $G$. The importance scores belonging to each group is defined as $\mathcal{I}_G = \{\mathcal{I}_G^{(i)}|g_i \in G\}$. For each group $g_i \in G$, importance scores are summed on every neuron over parameters of upper layers $L_u \subset g_i$ with length $N_u$ and lower layers $L_l \subset g_i$ with length $N_l$, as illustrated in in fig. 2.

**Unstructured Pruning** The importance score in unstructured learning is more straightforward. By eliminating the process of gathering importance, considering each parameter layer as a group, and each parameter as a neuron, the definition of importance score becomes similar to what is described in structured pruning scenarios. Experimentally we found that the gradients of some parameters are zero due to gradient vanishing, making us hard to judge the importance of these parameters. This would have little influence on structured pruning since neurons' importance scores are gathered through multiple weights. However, in unstructured pruning, the importance score of each weight only depends on itself. By adding the magnitude of the corresponding weight in the importance score term, the problem is resolved.

In each pruning step, we calculate every importance score $\mathcal{I}_G^{(i,j)}$ of $j^{th}$ neuron in $i^{th}$ group over training data of $B$ batches, and normalize them within each group. Then we rank them from lowerst to highest. The lowest $p$ percentage of parameters are pruned. We can gradually increase $p$ to iteratively prune the model to a specific FLOPs or parameter percentage.

## 5 RESULTS

We empirically study the performance of OBA on CNNs where only series connectivity exists, and attention networks where both series connectivity and parallel connectivity are present. We focus on pruning towards as small FLOPs as possible to reduce the computation overhead of model inference. For structured pruning, we primarily compare our methods with those that leverage Hessian matrix information (Hassibi & Stork, 1992; Wang et al., 2019). In the context of unstructured pruning, we evaluate our approach against the state-of-the-art unstructured pruning method, CHITA (Benbaki et al., 2023). Classical importance acquisition methods Weight (the magnitude of weights), OBD (LeCun et al., 1989) and Taylor (Molchanov et al., 2016) are also added into comparison for both structured and unstructured pruning tasks. In our experiments, we choose $\delta\theta_i = \theta_i$. The implementation details can be found at appendix A.

### 5.1 STRUCTURED PRUNING

Current structured pruning workflows can be roughly divided into one-shot pruning and iterative pruning. The former prunes the fine-tuned model towards a sparsified network and fine-tunes it after pruning, whereas the latter prunes the model iteratively and fine-tunes the model after each pruning step. One-shot pruning is more efficient than iterative pruning, but the latter is more effective. We evaluate our method on both of these two workflows.

### 5.1.1 IMPORTANCE SCORE RANK CHARACTERIZATION

We first measure the ability of our method to characterize the importance of each neuron for structured pruning. Intuitively, the importance score of a neuron should be positively correlated to the

Table 1: The Spearman correlation of the importance scores of each method with the importance rank of neurons on ResNet32.

| Method | OBA | Weight | Taylor | OBS | OBD | EigenDamage |
|---|---|---|---|---|---|---|
| *all layers* | | | **CIFAR100** | | | |
| Max | **0.500** | 0.115 | 0.457 | 0.248 | 0.222 | -0.077 |
| Standardization | **0.496** | 0.335 | 0.450 | 0.292 | 0.474 | -0.047 |
| $l_2$-Norm | 0.492 | 0.433 | 0.441 | 0.407 | **0.541** | -0.012 |
| None | **0.444** | -0.247 | -0.157 | 0.222 | 0.275 | 0.046 |
| *per layer* | **0.443** | 0.271 | 0.292 | 0.304 | 0.287 | 0.018 |
| *all layers* | | | **CIFAR10** | | | |
| Max | **0.456** | 0.092 | 0.417 | 0.373 | 0.426 | 0.165 |
| Standardization | 0.382 | 0.049 | **0.444** | 0.166 | 0.412 | 0.166 |
| $l_2$-norm | 0.402 | **0.443** | 0.433 | 0.367 | 0.186 | 0.192 |
| None | **0.379** | -0.187 | 0.080 | 0.360 | 0.342 | 0.172 |
| *per layer* | **0.421** | 0.289 | 0.409 | 0.362 | 0.347 | 0.056 |

Table 2: Comparison on Accuracies (%) and speed up up ratios with recent pruning works on ImageNet.

| Method | Pruned | Speed Up |
|---|---|---|
| **Resnet50** | | |
| Weight | $75.12_{-1.03}$ | $1.99\times$ |
| C-OBD (Wang et al., 2019) | $74.86_{-1.29}$ | $1.99\times$ |
| C-OBS (Wang et al., 2019) | $75.48_{-0.67}$ | $2.01\times$ |
| EigenDamage (Wang et al., 2019) | $75.30_{-0.85}$ | $2.00\times$ |
| Taylor (Molchanov et al., 2016) | $75.26_{-0.89}$ | $2.00\times$ |
| OBA(Ours) | $\mathbf{75.62}_{-0.53}$ | $2.00\times$ |
| **ViT-B/16** | | |
| Weight | $77.03_{-4.04}$ | $1.32\times$ |
| Taylor (Molchanov et al., 2016) | $77.65_{-3.42}$ | $1.31\times$ |
| OBA(Ours) | $\mathbf{79.64}_{-1.43}$ | $1.30\times$ |

change of loss when removing a neuron. This can be implemented by masking the neuron's output in the evaluation process. Specifically, we mask the output of each neuron in pre-defined layers and compare the change of loss between the masked model and unmasked model, which is refered to as ground truth importance. Then we calculate the Spearman's rank correlation of the importance scores with estimated importance scores from different methods. The higher the Spearman's rank score, the better the method captures the importance. We select the output neurons of the first convolutional layer and the three following residual blocks of ResNet32 (He et al., 2016) as candidates. For per-layer case, Spearman correlation is calculated within each layer and averaged. For all-layer case, before calculating Spearman correlation, we first normalize the importance scores of each layer and concatenate them together. Different layer-wise normalization methods are considered in the evaluation and are detailedly introduced in appendix C.

As shown in table 1, with evaluation within each layer, OBA yields the best rank similarity to the ground truth importance among all methods. In the all-layer importance condition, our method also yields good performance on different normalization methods. It is noteworthy that methods such as Weight, Taylor, and OBD, although potentially exhibiting higher correlation scores in importance under specific normalization schemes, may also yield notably low correlation scores under certain normalization techniques. In contrast, OBA consistently maintains relatively high correlation scores irrespective of the normalization method applied. This proves our method's capability of importance capturing, along with its robustness to different normalization methods.

### 5.1.2 ONE-SHOT PRUNING RESULTS

Next, we conduct experiments to evaluate the performance of OBA on pruning towards a specific FLOPs percentage in one-shot pruning workflow. As the FLOPs and number of parameters are not linearly dependent w.r.t. the number of neurons in the neural network, it's hard to calculate a pruning ratio under which the network is pruned into a predefined FLOPs or parameter number. Thus we prune the network for several steps with an increasing pruning ratio. FLOPs and number of parameters are calculated after each step to check whether the network satisfies the target. We first evaluate our method on ImageNet with ResNet50 (He et al., 2016) and ViT-B/16 (Dosovitskiy et al., 2020). As shown in table 5, our method realizes a $2\times$ speed up at ImageNet on ResNet50 with an accuracy decrease of only $0.53\%$. Our method outperforms other methods on both ResNet50 and ViT-B/16, which demonstrates the effectiveness of our method on large-scale datasets. On ViT-B/16, our method achieves a $1.30\times$ speed up with an accuracy decrease of $1.43\%$, which is far smaller than those achieved by Taylor criteria and Weight criteria, demonstrating our proposed criteria's superiority over Weight criteria and the first-order Taylor criteria.

Next we evaluate our method on CIFAR10 and CIFAR100 datasets. The results are shown in table 3. Our method achieves the best results on ResNet32 model across the both datasets. When pruning network towards a small $6\%$ target FLOPs, our method on ResNet32 surpasses other methods a lot, with only $0.79\%$ accuracy loss on CIFAR10 datasets. On CIFAR10 with VGG19 the performance of OBA is slightly worse, but still comparable to other methods, with relatively lower FLOPs. We observe an interesting phenomenon that the parameter reduction of models pruned with OBA are slightly higher than other methods under similar FLOPs, which indicates that our method tends to prune neurons with relatively higher FLOPs, and is more suitable for applications of lower FLOPs requirements.

Table 3: The accruacies (%), weights reduction (%), and FLOPs reduction (%) of different methods across CIFAR10 and CIFAR100 under one-shot pruning with ResNet32 and VGG19.

| Method | CIFAR10 | | | | | | CIFAR100 | | | | | |
|---|---|---|---|---|---|---|---|---|---|---|---|---|
| | Acc | Weights | FLOPs | Acc | Weights | FLOPs | Acc | Weights | FLOPs | Acc | Weights | FLOPs |
| **VGG19(Baseline)** | 94.17 | | | | | | 73.34 | | | | | |
| NN Slimming (Liu et al., 2017) | 92.84 | 80.07 | 42.65 | 85.01 | 97.85 | 97.89 | 71.89 | 74.60 | 38.33 | 58.69 | 97.76 | 94.09 |
| C-OBD (Wang et al., 2019) | 94.04 | 82.01 | 38.18 | 92.34 | 97.68 | 77.39 | 72.23 | 77.03 | 33.70 | 58.07 | 97.97 | 77.55 |
| C-OBS (Wang et al., 2019) | 94.08 | 76.96 | 34.73 | 91.92 | 97.27 | 87.53 | 72.27 | 73.83 | 38.09 | 58.87 | 97.61 | 91.94 |
| Kron-OBD (Wang et al., 2019) | 94.00 | 80.40 | 38.19 | **92.92** | 97.47 | 81.44 | 72.29 | 77.24 | 37.90 | 60.70 | 97.56 | 82.55 |
| Kron-OBS (Wang et al., 2019) | 94.09 | 79.71 | 36.93 | 92.56 | 97.32 | 80.39 | 72.12 | 74.18 | 36.59 | 60.66 | 97.48 | 83.57 |
| EigenDamage (Wang et al., 2019) | 93.98 | 78.18 | 37.13 | 92.29 | 97.15 | 86.51 | **72.90** | 76.64 | 37.40 | 65.18 | 97.31 | 88.63 |
| Weight | 93.85 | 68.57 | 37.21 | 91.85 | 97.02 | 86.93 | 72.14 | 67.89 | 37.02 | 54.63 | 95.90 | 88.30 |
| Taylor (Molchanov et al., 2016) | **94.11** | 62.29 | 38.28 | 92.29 | 93.89 | 86.29 | 71.82 | 55.58 | 37.76 | 66.65 | 93.12 | 88.28 |
| OBA (Ours) | 93.9 | 56.63 | 38.06 | 92.48 | 91.89 | 86.27 | 72.36 | 53.33 | 37.62 | **66.72** | 92.74 | 88.80 |
| **ResNet32(Baseline)** | 95.3 | | | | | | 78.17 | | | | | |
| C-OBD (Wang et al., 2019) | 95.11 | 70.36 | 66.18 | 91.75 | 97.30 | 93.50 | 75.70 | 66.68 | 67.53 | 59.52 | 97.74 | 94.88 |
| C-OBS (Wang et al., 2019) | 95.04 | 67.90 | 76.75 | 90.04 | 95.49 | 97.39 | 75.16 | 66.83 | 76.59 | 58.20 | 91.99 | 96.27 |
| Kron-OBD (Wang et al., 2019) | 95.11 | 63.97 | 63.41 | 92.57 | 96.11 | 94.18 | 75.86 | 63.92 | 62.97 | 62.42 | 96.42 | 95.85 |
| Kron-OBS (Wang et al., 2019) | 95.14 | 64.21 | 61.89 | 92.76 | 96.14 | 94.37 | 75.98 | 62.36 | 60.41 | 63.62 | 93.56 | 95.65 |
| EigenDamage (Wang et al., 2019) | 95.17 | 71.99 | 70.25 | 93.05 | 96.05 | 94.74 | 75.51 | 69.80 | 71.62 | 65.72 | 95.21 | 94.62 |
| Weight | 94.51 | 55.47 | 71.99 | 92.07 | 91.61 | 94.09 | 75.72 | 65.76 | 70.08 | 66.09 | 95.44 | 94.63 |
| Taylor (Molchanov et al., 2016) | 95.06 | 66.57 | 71.40 | 93.32 | 94.88 | 94.05 | 76.13 | 64.27 | 70.11 | 64.56 | 90.35 | 94.99 |
| OBA (Ours) | **95.19** | 63.90 | 71.13 | **93.45** | 93.68 | 94.51 | **76.47** | 64.34 | 70.02 | **67.81** | 94.43 | 94.40 |

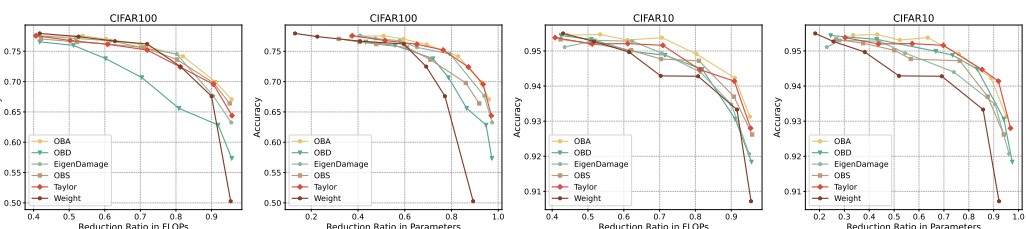

Figure 3: Iterative pruning results on CIFAR10 and CIFAR100 with ResNet32.

### 5.1.3 ITERATIVE PRUNING RESULTS

In this subsection, we evaluate the performance of OBA under iterative pruning workflow on ResNet32. Specifically, we set a list of target FLOPs for each dataset and model, and iteratively prune the model from largest FLOPs to smallest FLOPs. Fine-tuning is conducted after each pruning step. In terms of FLOPs, the accuracy loss for each iteration of our method is lower than other methods, as shown in fig. 3. This validates the effectiveness of our proposed criteria. As for parameter reduction, our method yields less advantage over other methods, which is consistent with the results of one-shot pruning.

### 5.2 UNSTRUCTURED PRUNING

We also evaluate the performance of OBA on unstructured pruning task. We follow a similar setting of the multi-stage pruning in CHITA (Benbaki et al., 2023) to have a fair comparison. Since Taylor (Molchanov et al., 2016), Weight, and OBD (LeCun et al., 1989) all obtain the importance scores from each parameter, we can ignore their importance gathering steps and add them into comparison. The results are shown in table 4 and table 5. Given the varying performance of the initial unpruned networks, we directly compare the accuracy ratio relative to the raw accuracy of all methods. The main version of CHITA, i.e. CHITA-CD, has very large computational time and memory cost on Resnet50, making itself infeasible for such huge networks. Thus we implemented the more efficient CHITA-BSO for comparison. It can be seen that Taylor and OBD fails on this task as their accuracies rapidly fall into $10\%$ in the first pruning step. OBA's results on high sparsities surpass CHITA++ by a huge margin, proving itself to be effective in unstructured pruning task.

### 5.3 RUN TIME ANALYSIS

Here, we empirically study the time consumption of OBA. Table 6a shows the time consumption of each part of OBA, and the time costed by regular training. It can be seen that the time cost of computing parallel connectivity is the most time-consuming part of OBA, nearly same to the time of series connectivity. In network structures that do not contain multiplication operations, the time cost of OBA would be much lower. In table 6b, 200 batches of data with a batch size of 64

Table 4: The unstructured pruning results on CIFAR-10 dataset with ResNet-20.

| Sparsity | Taylor (91%) | | OBD (91%) | | Weight (91%) | | OBA (91%) | | CHITA++ (91.36%) | |
|---|---|---|---|---|---|---|---|---|---|---|
| | Accuracy (%) | Ratio (%) | Accuracy (%) | Ratio (%) | Accuracy (%) | Ratio (%) | Accuracy (%) | Ratio (%) | Accuracy (%) | Ratio (%) |
| 0.1 | 11.00 | 14.45 | 10.03 | 13.17 | 90.66 | 99.63 | 90.83 | **99.81** | - | - |
| 0.2 | 11.00 | 14.45 | 10.03 | 13.17 | 90.82 | 99.80 | 90.90 | **99.89** | - | - |
| 0.3 | 10.00 | 13.14 | 10.03 | 13.17 | 90.67 | 99.64 | 90.65 | 99.62 | 91.25 | **99.88** |
| 0.4 | 10.80 | 14.19 | 10.02 | 13.16 | 90.67 | 99.64 | 90.35 | 99.29 | 91.20 | **99.82** |
| 0.5 | 10.00 | 13.14 | 10.01 | 13.15 | 90.79 | **99.77** | 90.57 | 99.53 | 91.04 | 99.65 |
| 0.6 | 8.20 | 10.77 | 10.00 | 13.14 | 90.23 | 99.15 | 90.69 | **99.66** | 90.78 | 99.37 |
| 0.7 | 10.00 | 13.14 | 10.00 | 13.14 | 88.83 | 97.62 | 89.94 | 98.84 | 90.38 | **98.93** |
| 0.8 | 10.00 | 13.14 | 10.00 | 13.14 | 85.03 | 93.44 | 89.64 | **98.51** | 88.72 | 97.11 |
| 0.9 | 10.00 | 13.14 | 10.52 | 13.82 | 67.00 | 73.63 | 86.27 | **94.80** | 79.32 | 86.82 |

Table 5: The unstructured pruning results on Imagenet dataset with ResNet-50.

| Sparsity | Taylor (76.13%) | | OBD (76.13%) | | Weight (76.13%) | | OBA (76.13%) | | CHITA-BSO++ (77.01%) | |
|---|---|---|---|---|---|---|---|---|---|---|
| | Accuracy (%) | Ratio (%) | Accuracy (%) | Ratio (%) | Accuracy (%) | Ratio (%) | Accuracy (%) | Ratio (%) | Accuracy (%) | Ratio (%) |
| 0.1 | 0.11 | 0.14 | 0.10 | 0.13 | 75.30 | 98.91 | 75.67 | 99.40 | 77.00 | **99.98** |
| 0.2 | 0.11 | 0.14 | 0.10 | 0.13 | 75.23 | 98.82 | 75.40 | 99.04 | 76.91 | **99.87** |
| 0.3 | 0.10 | 0.13 | 0.10 | 0.13 | 74.57 | 97.95 | 74.93 | 98.42 | 76.87 | **99.82** |
| 0.4 | 0.11 | 0.14 | 0.10 | 0.13 | 73.54 | 96.60 | 74.50 | 97.86 | 76.59 | **99.46** |
| 0.5 | 0.10 | 0.13 | 0.10 | 0.13 | 70.73 | 92.91 | 73.42 | 96.44 | 76.01 | **98.70** |
| 0.6 | 0.08 | 0.11 | 0.10 | 0.13 | 64.97 | 85.34 | 70.88 | **93.10** | 68.89 | 89.46 |
| 0.7 | 0.10 | 0.13 | 0.10 | 0.13 | 48.09 | 63.17 | 65.57 | **86.13** | 64.38 | 83.60 |
| 0.8 | 0.10 | 0.13 | 0.09 | 0.12 | 16.08 | 21.12 | 47.14 | **61.92** | 26.21 | 34.03 |
| 0.9 | 0.10 | 0.13 | 0.10 | 0.13 | 0.80 | 1.05 | 5.65 | **7.43** | 0.43 | 0.56 |

| | Resnet32 | ViT-B/16 |
|---|---|---|
| Regular Traning | 0.326 | 0.252 |
| Upper Series | 0.881 | 0.799 |
| Lower Series | 0.819 | 1.166 |
| Parallel | - | 1.944 |
| Total | 2.072 | 4.027 |

(a)

(b)

(c)

Table 6: The running time (s) of OBA on different models and datasets. **(a)** The specific running time (s/iteration) of each part of OBA on ResNet32 and ViT-B/16. **(b)** The running time of OBA on ResNet32 with CIFAR100. **(c)** The running time of OBA on ResNet20 with CIFAR10.

are leveraged in each pruning iteration. In each pruning stage we conduct 50 pruning iterations to gradually prune the network, and the overall computation time of pruning would be around 4.2 hours, which is acceptable in the actual pruning scenarios. In table 6c, we samely use 200 batches of data to calculate gradients. The mainly proposed method in CHITA, CHITA-CD, would require twice as much time compared to our approach. However, the performance of both CHITA-CD and CHITA-BSO under high sparsity is worse than OBA, showcasing the efficiency of our method.

## 6 CONCLUSION AND LIMITATION

In this paper, we propose Optimal Brain Apoptosis, a novel method for pruning neural networks. We first provide theoretical analysis on modern neural network structures to figure out nonzero Hessian submatrix conditions between layers. Then we propose an efficient approach that directly calculates the Hessian-vector product values for each parameter in the network, thereby calculating the second-order Taylor expansion for each parameter without any approximation. We empirically demonstrate the efficacy of our method on both structured pruning and unstructured pruning.

**Limitation** OBA, in its current form, can be applied to network structures including MLPs, CNNs, and Transformers. For networks with more complex architectures, like RNNs and State Space Models that handle time-series data, computing the Hessian matrix becomes more difficult and necessitates additional research. This is an interesting area that warrants further exploration in the future.

## 7 ACKNOWLEDGEMENT

We sincerely appreciate the dedication and valuable feedback provided by all anonymous reviewers. This research was partially supported by the National Natural Science Foundation of China under Grants 62073066 and 62405255, as well as the Fundamental Research Funds for the Central Universities under Grant N2226001. Additionally, it received support from the 111 Project under Grant B16009 and the Intel Neuromorphic Research Community (INRC) Grant Award (RV2.137.Fang).

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

# APPENDIX

## A    IMPLEMENTATION DETAILS

In our structured pruning experiments, we align our settings to EigenDamage (Wang et al., 2019), which is a good Hessian based pruning method. For each pruning step, we obtain the importance score from 200 batches of data to calculate the importance score. We set the batch size to 64 and the learning rate to 0.001 for the fine-tuning process. We use the SGD optimizer with a momentum of 0.9 and a weight decay of $5 \times 10^{-4}$. The learning rate is divided by 10 at the 80th and 120th epochs. We set the maximum epochs to 150 for CIFAR10 and CIFAR100 datasets. For ImageNet experiments, we use the same settings as Fang et al. (2023) of ResNet50 and ViT-B/16. We set the maximum epochs to 100 for ImageNet experiments.

---

**Algorithm 1** Importance Score Acquisition of OBA

---

**Input:**  model $m$ and its parameters $\theta$, a batch of data $\mathcal{D}$
**Output:**  parameter importance $\mathcal{I}$
1: Initialized importance dict $\mathcal{I}$
2: Conduct one forward propagation and back propagation process on model $m$ with data $\mathcal{D}$ and record the output gradient $\frac{\partial \mathcal{L}}{\partial y^{(l)}}$ for each layer $l$
3: **for** parameter layer $l$ in $m$ **do**
4:     $x^{(l)}$.backward($\frac{\partial \mathcal{L}}{\partial y^{(l)}} \mathbf{J}^{(l)}_{\delta W^{(l)}}$)
5: **end for**
6: **for** parameter layer $l$ in $m$ **do**
7:     $\mathcal{I}[l] \leftarrow \delta\theta^{(l)} \odot \theta^{(l)}$.grad
8: **end for**
9: Conduct JVPF according to eq. (14) and record $\hat{X}^{(l)}$ for each layer $l$
10: $m$.zero_grad()
11: **for** parameter layer $l$ in $m$ **do**
12:     $\mathcal{I}[l] \leftarrow \mathcal{I}[l] + \frac{\partial \mathcal{L}}{\partial y^{(l)}} \frac{\partial y^{(l)}}{\partial \theta^{(l)}}\big|_{\hat{X}^{(l)}} \odot \delta\theta^{(l)}$
13:     **if** layer $l$ is attention module **then**
14:         $\hat{S} = \mathrm{softmax}\big((Q^{(l)}\hat{K}^{(l)\mathsf{T}} + \hat{Q}^{(l)}K^{(l)\mathsf{T}})/\sqrt{d_k^{(l)}}\big)$
15:         $S = \mathrm{softmax}\big(Q^{(l)}K^{(l)\mathsf{T}}/\sqrt{d_k^{(l)}}\big)$
16:         $\hat{O} = \hat{S}V + S\hat{V}, O = SV$
17:         $\hat{O}$.backward($\frac{\partial \mathcal{L}}{\partial O}$)
18:     **end if**
19: **end for**
20: **for** parameter layer $l$ in $m$ **do**
21:     $\mathcal{I}[l] \leftarrow \mathcal{I}[l] + \delta\theta^{(l)} \odot \theta^{(l)}$.grad
22: **end for**
23: $m$.zero_grad()

---

## B    ALGORITHMIC DETAILS

As can be seen in algorithm 1, We first conduct a forward propagation process on the network and record the output gradient of the loss w.r.t. the output of each layer (line 1-2). Then we back-propagate these gradients for each parameter layer with corresponding weight $\delta\theta$ to obtain the term $\sum_{l' \in \mathbf{L}_{\mathrm{up}}} \sum_j \frac{\partial^2 \mathcal{L}}{\partial \theta_j^{(l')} \partial \theta^{(l)}} \delta\theta_j^{(l')}$ (line 3-5).

1. **Upper Series Connectivity:** These terms are recorded in the gradient of the corresponding parameters, so that we can obtain $\sum_{l' \in \mathbf{L}_{\mathrm{up}}} \sum_j \frac{\partial^2 \mathcal{L}}{\partial \theta_j^{(l')} \partial \theta^{(l)}} \delta\theta_j^{(l')} \odot \delta\theta^{(l)}$ in eq. (6) by multiplying the parameters with their gradients (line 6-7).

2. Next, We obtain $\hat{X}_{mn}^{(l)} = \sum_{l_{\text{low}} \in \mathbf{L}_{\text{low}}} \sum_k \frac{\partial X_{mn}^{(l)}}{\partial \theta_k^{(l_{\text{low}})}} \delta \theta_k^{(l_{\text{low}})}$ for all parameter layers through the JVPF, these values are useful for calculating **Lower Series Connectivity** cases and **Parallel Connectivity** cases (line 9).

3. **Lower Series Connectivity:** We obtain the latter term of eq. (6) and add them into the importance scores (line 12).

4. **Parallel Connectivity:** In the meantime, for all attention layers that induce parallel connectivity, we back-propagate the gradient with the surrogate inputs $\hat{X}_{\text{left}}$ and $\hat{X}_{\text{right}}$ according to eqs. (12) and (13) (line 13-17), and multiply the parameters with their gradients that are in **Lower Series Connectivity** with the attention layer (line 20-21).

## C NORMALIZATION METHODS

In our implementation, We leverage these normalization methods on importance scores for each group, including:

**No Normalization (None)** When the normalizer is set to None, the original values of importance scores are returned without any modification. This means that the data is used as-is, with all its original properties (such as scale and distribution) intact:

$$I_j^{\text{normalized}} = I_j.$$

**Standardization (or Min-Max Normalization)** This method scales the data so that it fits within a specific range, typically 0 to 1. This is achieved by subtracting the minimum value of the data and then dividing by the range of the data:

$$I_j^{\text{normalized}} = \frac{I_j - \min(I)}{\max(I) - \min(I)}.$$

**Max Normalization** In this approach, every importance score is divided by the maximum importance score of corresponding group to ensure that all the normalized values fall between 0 and 1:

$$I_j^{\text{normalized}} = \frac{I_j}{\max(I)}.$$

$l_2$ **Normalization** This method normalizes the importance scores by dividing it by the $l_2$ norm (Euclidean norm) of the importance scores belonging to the same group. The $l_2$ norm is calculated as the square root of the sum of the squared values:

$$I_j^{\text{normalized}} = \frac{I_j}{||I||_2}.$$

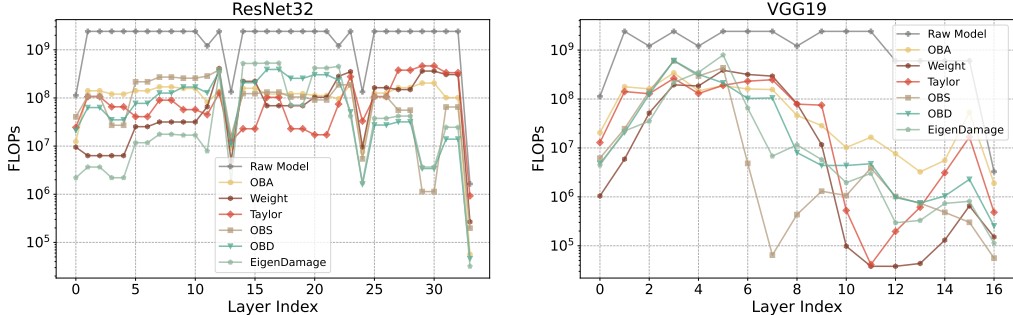

Figure A1: The layer-wise FLOPs for all parameter layers from models pruned by different criteria.

## D  PRUNED LAYERS VISUALIZATION

We visualized the layer-wise FLOPs of pruned models by OBA and other methods for ResNet32 and VGG19 on CIFAR100 with a target FLOPs of $6\%$. It can be seen that compared with other methods, the difference of OBA between FLOPs of different layers is smaller, resulting in a smoother model in terms of number of neurons across layers. This significantly helps to improve the model's performance across various datasets and provide useful guidance for researchers to design and prune neural networks.

## E  PROOFS

### E.1  THEOREM 4.2

For any two layers $l_{\text{low}}$ and $l_{\text{up}}$ in series connectivity, where $l_{\text{up}}$ is upper than $l_{\text{low}}$. Note that $\frac{\partial^2 \mathcal{L}}{\partial w^{(l_{\text{low}})} \partial b^{(l_{\text{up}})}}$ is a zero matrix because parameters $w^{(l_{\text{low}})}$ and $b^{(l_{\text{up}})}$ are independent of each other. With this prior, $\frac{\partial^2 \mathcal{L}}{\partial \theta_i^{(l_{\text{low}})} \partial \theta_j^{(l_{\text{up}})}}$ is actually $\frac{\partial^2 \mathcal{L}}{\partial \theta_i^{(l_{\text{low}})} \partial w_j^{(l_{\text{up}})}}$. We first calculate term $\sum_j \frac{\partial^2 \mathcal{L}}{\partial \theta_i^{(l_{\text{low}})} \partial w_j^{(l_{\text{up}})}} \delta \theta_i^{(l_{\text{low}})} \delta w_j^{(l_{\text{up}})}$ for the lower layer $l_{\text{low}}$.

**Lemma E.1.**

$$\sum_{fhi} \frac{\partial^2 \mathcal{L}}{\partial W_{fhi}^{(l_{\text{up}})} \partial W_{acd}^{(l_{\text{low}})}} \delta W_{fhi}^{(l_{\text{up}})} \delta W_{acd}^{(l_{\text{low}})} = \frac{\partial \mathcal{L}}{\partial y^{(l_{\text{up}})}} \mathbf{J}_{\delta W^{(l_{\text{up}})}}^{(l_{\text{up}})} \frac{\partial x^{(l_{\text{up}})}}{\partial W_{acd}^{(l_{\text{low}})}} \delta W_{acd}^{(l_{\text{low}})} \tag{15}$$

*and*

$$\sum_{fhi} \frac{\partial^2 \mathcal{L}}{\partial W_{fhi}^{(l_{\text{up}})} \partial b_a^{(l_{\text{low}})}} \delta W_{fhi}^{(l_{\text{up}})} \delta b_a^{(l_{\text{low}})} = \frac{\partial \mathcal{L}}{\partial y^{(l_{\text{up}})}} \mathbf{J}_{\delta W^{(l_{\text{up}})}}^{(l_{\text{up}})} \frac{\partial x^{(l_{\text{up}})}}{\partial b_a^{(l_{\text{low}})}} b_a^{(l_{\text{low}})}, \tag{16}$$

*where $\mathbf{J}_{\delta W^{(l_{\text{up}})}}^{(l_{\text{up}})} \in \mathbb{R}^{(l_{out} \cdot m_{out}) \times (l_{in} \cdot m_{in})}$ is the jacobian matrix of $y^{(l_{\text{up}})}$ with respect to $x^{(l_{\text{up}})}$ taking $\delta W^{(l_{\text{up}})}$ as the weights.*

*Proof.* Let $\mathbf{J}_{W^{(l_{\text{up}})}}^{(l_{\text{up}})} = \frac{\partial y^{(l_{\text{up}})}}{\partial x^{(l_{\text{up}})}}$. The element-wise derivative of loss w.r.t. $X^{(l_{\text{up}})}$ is given by

$$\frac{\partial \mathcal{L}}{\partial X_{hj}^{(l_{\text{up}})}} = \sum_{fg} \frac{\partial \mathcal{L}}{\partial Y_{fg}^{(l_{\text{up}})}} \underbrace{\sum_{ij} W_{fhi}^{(l_{\text{up}})} M_{gij}^{(l_{\text{up}})}}_{\partial Y_{fg}^{(l_{\text{up}})} / \partial X_{hj}^{(l_{\text{up}})}}, \tag{17}$$

and the element-wise derivative of loss w.r.t. $W^{(l_{\text{up}})}$ is given by

$$\frac{\partial \mathcal{L}}{\partial W_{fhi}^{(l_{\text{up}})}} = \sum_g \frac{\partial \mathcal{L}}{\partial Y_{fg}^{(l_{\text{up}})}} \sum_j X_{hj}^{(l_{\text{up}})} M_{gij}^{(l_{\text{up}})}. \tag{18}$$

Applying chain rule, we have

$$\sum_{fhi} \frac{\partial^2 \mathcal{L}}{\partial W_{fhi}^{(l_{\text{up}})} \partial W_{acd}^{(l_{\text{low}})}} \delta W_{fhi}^{(l_{\text{up}})} \delta W_{acd}^{(l_{\text{low}})} = \sum_{fg} \frac{\partial \mathcal{L}}{\partial Y_{fg}^{(l_{\text{up}})}} \sum_{hij} M_{gij}^{(l_{\text{up}})} \delta W_{fhi}^{(l_{\text{up}})} \frac{\partial X_{hj}^{(l_{\text{up}})}}{\partial W_{acd}^{(l_{\text{low}})}} \delta W_{acd}^{(l_{\text{low}})}$$

$$= \frac{\partial \mathcal{L}}{\partial y^{(l_{\text{up}})}} \mathbf{J}_{\delta W^{(l_{\text{up}})}}^{(l_{\text{up}})} \frac{\partial x^{(l_{\text{up}})}}{\partial W_{acd}^{(l_{\text{low}})}} \delta W_{acd}^{(l_{\text{low}})}, \tag{19}$$

and

$$\sum_{fhi} \frac{\partial^2 \mathcal{L}}{\partial W_{fhi}^{(l_{\text{up}})} \partial b_a^{(l_{\text{low}})}} \delta W_{fhi}^{(l_{\text{up}})} \delta b_a^{(l_{\text{low}})} = \sum_{fg} \frac{\partial \mathcal{L}}{\partial Y_{fg}^{(l_{\text{up}})}} \sum_{hij} M_{gij}^{(l_{\text{up}})} \delta W_{fhi}^{(l_{\text{up}})} \frac{\partial X_{hj}^{(l_{\text{up}})}}{\partial b_a^{(l_{\text{low}})}} b_a^{(l_{\text{low}})}$$

$$= \frac{\partial \mathcal{L}}{\partial y^{(l_{\text{up}})}} \mathbf{J}_{\delta W^{(l_{\text{up}})}}^{(l_{\text{up}})} \frac{\partial x^{(l_{\text{up}})}}{\partial b_a^{(l_{\text{low}})}} b_a^{(l_{\text{low}})}. \tag{20}$$

$\square$

Next we obtain term $\sum_i \frac{\partial^2 \mathcal{L}}{\partial \theta_i^{(l_{\text{low}})} \partial w_j^{(l_{\text{up}})}} \delta\theta_i^{(l_{\text{low}})} \delta w_j^{(l_{\text{up}})}$ for the upper layer $l_{\text{up}}$.

**Lemma E.2.**

$$\sum_a \frac{\partial^2 \mathcal{L}}{\partial W_{fhi}^{(l_{up})} \partial \theta_a^{(l_{low})}} \delta W_{fhi}^{(l_{up})} \delta\theta_a^{(l_{low})} = \frac{\partial \mathcal{L}}{\partial y^{(l_{up})}} \frac{\partial y^{(l_{up})}}{\partial W_{fhi}^{(l_{up})}} \Bigg|_{\hat{X}^{(l_{up},l_{low})}} \delta W_{fhi}^{(l_{up})}, \tag{21}$$

where $\hat{X}^{(l_{up},l_{low})}$ is given by

$$\hat{X}_{hj}^{(l_{up},l_{low})} = \sum_a \frac{\partial X_{hj}^{(l_{up},l_{low})}}{\partial \theta_a^{(l_{low})}} \delta\theta_a^{(l_{low})}. \tag{22}$$

*Proof.*

$$\frac{\partial \mathcal{L}}{\partial y^{(l_{up})}} \frac{\partial y^{(l_{up})}}{\partial W_{fhi}^{(l_{up})}} \Bigg|_{\hat{X}^{(l_{up})}} \delta W_{fhi}^{(l_{up})} = \sum_{fg} \frac{\partial \mathcal{L}}{\partial Y_{fg}^{(l_{up})}} \sum_j M_{gij}^{(l_{up})} \sum_a \frac{\partial X_{hj}^{(l_{up})}}{\partial \theta_a^{(l_{low})}} \delta\theta_a^{(l_{low})} \delta W_{fhi}^{(l_{up})}$$

$$= \sum_a \sum_g \frac{\partial \mathcal{L}}{\partial Y_{fg}^{(l_{up})}} \sum_j M_{gij}^{(l_{up})} \frac{\partial X_{hj}^{(l_{up})}}{\partial \theta_a^{(l_{low})}} \delta\theta_a^{(l_{low})} \delta W_{fhi}^{(l_{up})}$$

$$= \sum_a \frac{\partial^2 \mathcal{L}}{\partial W_{fhi}^{(l_{up})} \partial \theta_a^{(l_{up})}} \delta\theta_a^{(l_{low})} \delta W_{fhi}^{(l_{up})}. \tag{23}$$

$\square$

*Proof of theorem 4.2.* Sum the results in lemma E.1 for upper series connectivity cases with results in lemma E.2 for lower series connectivity cases and we get

$$\sum_{l' \in \mathbf{L}_{\text{up}} \cup \mathbf{L}_{\text{low}}} \sum_j \frac{\partial^2 \mathcal{L}}{\partial \theta_j^{(l')} \partial w_i^{(l)}} \delta\theta_j^{(l')} \delta w_i^{(l)} = \sum_{l_{\text{up}} \in \mathbf{L}_{\text{up}}} \frac{\partial \mathcal{L}}{\partial y^{(l_{\text{up}})}} \mathbf{J}_{\delta W^{(l_{\text{up}})}}^{(l_{\text{up}})} \frac{\partial x^{(l_{\text{up}})}}{\partial w_i^{(l)}} \delta w_i^{(l)} + \frac{\partial \mathcal{L}}{\partial y^{(l)}} \frac{\partial y^{(l)}}{\partial w_i^{(l)}} \Bigg|_{\hat{X}^{(l)}} \delta w_i^{(l)} \tag{24}$$

with

$$\hat{X}_{hj}^{(l)} = \sum_{l_{\text{low}} \in \mathbf{L}_{\text{low}}} \hat{X}_{hj}^{(l,l_{\text{low}})} = \sum_{l_{\text{low}} \in \mathbf{L}_{\text{low}}} \sum_k \frac{\partial X_{hj}^{(l)}}{\partial \theta_k^{(l_{\text{low}})}} \delta\theta_k^{(l_{\text{low}})},$$

and

$$\sum_{l' \in \mathbf{L}_{\text{up}} \cup \mathbf{L}_{\text{low}}} \sum_j \frac{\partial^2 \mathcal{L}}{\partial \theta_j^{(l')} \partial b_i^{(l)}} \delta\theta_j^{(l')} \delta b_i^{(l)} = \sum_{l_{\text{up}} \in \mathbf{L}_{\text{up}}} \frac{\partial \mathcal{L}}{\partial y^{(l_{\text{up}})}} \mathbf{J}_{\delta W^{(l_{\text{up}})}}^{(l_{\text{up}})} \frac{\partial x^{(l_{\text{up}})}}{\partial b_i^{(l)}} \delta b_i^{(l)}. \tag{25}$$

Rewrite eq. (24) and eq. (25) as vector forms and we obtain eq. (6) and eq. (9). $\square$

Appendix

### E.2 THEOREM 4.4

Here we follow the notations in definition 4.3 to denote each value of the two layers.

**Lemma E.3.** *Let $\hat{X}^{(left)} \in \mathbb{R}^{l_{row} \times l_{hid}}$ and $\hat{X}^{(right)} \in \mathbb{R}^{l_{hid} \times l_{col}}$ be two surrogate weight matrices such that*

$$\hat{X}_{kn}^{(left)} = \sum_q \frac{\partial X_{kn}^{(left)}}{\partial \theta_q^{(l_l)}} \delta\theta_q^{(l_l)}, \tag{26}$$

$$\hat{X}_{no}^{(right)} = \sum_r \frac{\partial X_{no}^{(right)}}{\partial \theta_r^{(l_r)}} \delta\theta_r^{(l_r)}. \tag{27}$$

*Then we have*

$$\sum_q \frac{\partial^2 \mathcal{L}}{\partial \theta_r^{(l_r)} \partial \theta_q^{(l_l)}} \delta\theta_r^{(l_r)} \delta\theta_q^{(l_l)} = \left. \frac{\partial \mathcal{L}}{\partial y^{(mul)}} \frac{\partial y^{(mul)}}{\partial x^{(right)}} \right|_{\hat{X}^{(left)}} \frac{\partial x^{(right)}}{\partial \theta_r^{(l_r)}} \delta\theta_r^{(l_r)} \tag{28}$$

*and*

$$\sum_r \frac{\partial^2 \mathcal{L}}{\partial \theta_r^{(l_r)} \partial \theta_q^{(l_l)}} \delta\theta_r^{(l_r)} \delta\theta_q^{(l_l)} = \left. \frac{\partial \mathcal{L}}{\partial y^{(mul)}} \frac{\partial y^{(mul)}}{\partial x^{(left)}} \right|_{\hat{X}^{(right)}} \frac{\partial x^{(left)}}{\partial \theta_q^{(l_l)}} \delta\theta_q^{(l_l)}. \tag{29}$$

*Proof.* The second-order partial derivative of loss w.r.t. $\theta_r^{(l_r)}$ and $\theta_q^{(l_l)}$ can be written as

$$\frac{\partial^2 \mathcal{L}}{\partial \theta_r^{(l_r)} \partial \theta_q^{(l_l)}} = \sum_{ko} \frac{\partial \mathcal{L}}{\partial Y_{ko}^{(mul)}} \sum_{np} \frac{\partial^2 Y_{ko}^{(mul)}}{\partial X_{kn}^{(left)} \partial X_{po}^{(right)}} \frac{\partial X_{po}^{(right)}}{\partial \theta_r^{(l_r)}} \frac{\partial X_{kn}^{(left)}}{\partial \theta_q^{(l_l)}}. \tag{30}$$

Since the multiplication can be expressed as $Y_{ko}^{(mul)} = \sum_n X_{kn}^{(left)} X_{no}^{(right)}$, which means

$$\frac{\partial^2 Y_{ko}^{(mul)}}{\partial X_{kn}^{(left)} \partial X_{po}^{(right)}} = \begin{cases} 1 & n = p, \\ 0 & n \neq p. \end{cases} \tag{31}$$

With this, eq. (30) can be rewritten as

$$\frac{\partial^2 \mathcal{L}}{\partial \theta_r^{(l_r)} \partial \theta_q^{(l_l)}} = \sum_{ko} \frac{\partial \mathcal{L}}{\partial Y_{ko}^{(mul)}} \sum_n \frac{\partial X_{no}^{(right)}}{\partial \theta_r^{(l_r)}} \frac{\partial X_{kn}^{(left)}}{\partial \theta_q^{(l_l)}}. \tag{32}$$

By expanding the right-hand side of eq. (28) we have

$$\sum_{ko} \frac{\partial \mathcal{L}}{\partial Y_{ko}^{(mul)}} \sum_n \frac{\partial X_{no}^{(right)}}{\partial \theta_r^{(l_r)}} \delta\theta_r^{(l_r)} \sum_q \frac{\partial X_{kn}^{(left)}}{\partial \theta_q^{(l_l)}} \delta\theta_q^{(l_l)} \tag{33}$$

$$= \sum_q \sum_{ko} \frac{\partial \mathcal{L}}{\partial Y_{ko}^{(mul)}} \sum_n \frac{\partial X_{no}^{(right)}}{\partial \theta_r^{(l_r)}} \frac{\partial X_{kn}^{(left)}}{\partial \theta_q^{(l_l)}} \delta\theta_r^{(l_r)} \delta\theta_q^{(l_l)}$$

$$= \sum_q \frac{\partial^2 \mathcal{L}}{\partial \theta_r^{(l_r)} \partial \theta_q^{(l_l)}} \delta\theta_r^{(l_r)} \delta\theta_q^{(l_l)}. \tag{34}$$

Expand the right-hand side of eq. (29) and we can see it also holds. $\qquad\square$

*Proof of theorem 4.4.* According to lemma E.3, we have

$$\sum_{l_l \in \mathbf{L}_{\text{left}}} \sum_q \frac{\partial^2 \mathcal{L}}{\partial \theta_r^{(l_r)} \partial \theta_q^{(l_l)}} \delta\theta_r^{(l_r)} \delta\theta_q^{(l_l)} = \sum_{l_l \in \mathbf{L}_{\text{left}}} \sum_q \sum_{ko} \frac{\partial \mathcal{L}}{\partial Y_{ko}^{(mul)}} \sum_n \frac{\partial X_{no}^{(right)}}{\partial \theta_r^{(l_r)}} \frac{\partial X_{kn}^{(left)}}{\partial \theta_q^{(l_l)}} \delta\theta_r^{(l_r)} \delta\theta_q^{(l_l)}$$

$$= \sum_{ko} \frac{\partial \mathcal{L}}{\partial Y_{ko}^{(mul)}} \sum_n \frac{\partial X_{no}^{(right)}}{\partial \theta_r^{(l_r)}} \delta\theta_r^{(l_r)} \sum_{l_l \in \mathbf{L}_{\text{left}}} \sum_q \frac{\partial X_{kn}^{(left)}}{\partial \theta_q^{(l_l)}} \delta\theta_q^{(l_l)}$$

$$= \left. \frac{\partial \mathcal{L}}{\partial y^{(mul)}} \frac{\partial y^{(mul)}}{\partial x^{(right)}} \right|_{\sum_{l_l \in \mathbf{L}_{\text{left}}} \sum_q \frac{\partial X_{kn}^{(left)}}{\partial \theta_q^{(l_l)}} \delta\theta_q^{(l_l)}} \frac{\partial x^{(right)}}{\partial \theta_r^{(l_r)}} \delta\theta_r^{(l_r)}. \tag{35}$$

Apply similar operations on $\sum_{l_r \in \mathbf{L}_{\text{right}}} \sum_r \frac{\partial^2 \mathcal{L}}{\partial \theta_r^{(l_r)} \partial \theta_q^{(l_l)}} \delta\theta_r^{(l_r)} \delta\theta_q^{(l_l)}$ and we can get eq. (13). $\qquad\square$