# OpenReview forum: "Optimal Brain Apoptosis"
_ICLR.cc/2025/Conference — ICLR 2025 Poster_

### Official Review · Reviewer_fJfY · 2024-11-03

**Soundness:** 3
**Presentation:** 2
**Contribution:** 2
**Rating:** 5
**Confidence:** 4

**Summary:**

The paper concerns the importance of pruning to reduce computational burden of neural network inference. The structure of the Hessian matrix is analysed with a productive derivation of structure for  feedforward nets with so-called series and parallel connectivity. Empirical results are presented for standard (vision) CNNs.

**Strengths:**

The paper’s most prominent strength is the set of Hessian structures derived for series and parallel connectivity. The empirical results are promising showing significant reductions in FLOPS for virtually unchanged performance.

**Weaknesses:**

1)	The field is very busy with many results going back to the early 90s. Computational results and simplifying structure of the Hessian for feedforward network are found in many papers including
•	Wille, J., 1997, June. On the structure of the Hessian matrix in feedforward networks and second derivative methods. In Proceedings of International Conference on Neural Networks (ICNN'97) (Vol. 3, pp. 1851-1855). IEEE.
•	Buntine, W.L. and Weigend, A.S., 1994. Computing second derivatives in feed-forward networks: A review. IEEE transactions on Neural Networks, 5(3), pp.480-488.
•	Wu, Y., Zhu, X., Wu, C., Wang, A. and Ge, R., 2020. Dissecting hessian: Understanding common structure of hessian in neural networks. arXiv preprint arXiv:2010.04261
•	Singh, S.P., Bachmann, G. and Hofmann, T., 2021. Analytic insights into structure and rank of neural network hessian maps. Advances in Neural Information Processing Systems, 34, pp.23914-23927.

The paper presents a reduced complexity calculation of Hessian x vector, this is well-known - see work by Barak Pearlmutter for example; Pearlmutter, B.A., 1994. Fast exact multiplication by the Hessian. Neural computation, 6(1), pp.147-160.

•	In general the context of related work is not sufficient. I am not a fan of placing context related work in an appendix.

**Questions:**

1)	Empirical evidence: While the results for CNNs are impressive, please consider transformers. There is much interest in pruning and redundancy in transformers e.g.
a.	Men, X., Xu, M., Zhang, Q., Wang, B., Lin, H., Lu, Y., Han, X. and Chen, W., 2024. Shortgpt: Layers in large language models are more redundant than you expect. arXiv preprint arXiv:2403.03853.
b.	Lad, V., Gurnee, W. and Tegmark, M., 2024. The Remarkable Robustness of LLMs: Stages of Inference?. arXiv preprint arXiv:2406.19384.

2)	L652 Consider setting the scene, motivation and related work as an integral part of the paper. With due refence to early work on pruning and Hessian structure

3)	L058-9 The OBS method is only different from OBD when the non-diagonal Hessian is used, i.e they do not ignore the off-diagonal terms in OBS

4)	L095 You mention that the Gauss.Newton approximation is in-sufficient what is the evidence?

5)	L087 Do you keep track of the first order term (which is assumed zero in OBD/OBS)?

---

> ### Author Response · Authors · 2024-11-20
> **Reply to Reviewer fJfY (1/2)**
>
> We appreciate the valuable suggestions and efforts from the reviewer!
>
> > **Weakness 1** The field is very busy with many results going back to the early 90s. Computational results and simplifying structure of the Hessian for feedforward network are found in many papers including • Wille, J., 1997, June. On the structure of the Hessian matrix in feedforward networks and second derivative methods. In Proceedings of International Conference on Neural Networks (ICNN'97) (Vol. 3, pp. 1851-1855). IEEE. • Buntine, W.L. and Weigend, A.S., 1994. Computing second derivatives in feed-forward networks: A review. IEEE transactions on Neural Networks, 5(3), pp.480-488. • Wu, Y., Zhu, X., Wu, C., Wang, A. and Ge, R., 2020. Dissecting hessian: Understanding common structure of hessian in neural networks. arXiv preprint arXiv:2010.04261 • Singh, S.P., Bachmann, G. and Hofmann, T., 2021. Analytic insights into structure and rank of neural network hessian maps. Advances in Neural Information Processing Systems, 34, pp.23914-23927.
>
> Thanks for pointing out these literatures that build the ground of the Hessian in the neural network.
> We've added them to our related work, merging the Hessian Matrix part into section 2 (preliminary) as part of the main paper. For more details please refer to section 2 of our revised manuscript.
>
> > **Weakness 2** The paper presents a reduced complexity calculation of Hessian x vector, this is well-known - see work by Barak Pearlmutter for example; Pearlmutter, B.A., 1994. Fast exact multiplication by the Hessian. Neural computation, 6(1), pp.147-160.
>
> We are very grateful for your introduction to this previously unknown work! While our research shares a
> fundamental concept with "Fast Exact Multiplication by the Hessian," the specifics of our approaches
> differ significantly. The work in 1994 introduces a differential operator,
> $\mathcal{R}v({f(\boldsymbol{w})}) = \left.(\partial / \partial r) f(\mathbf{w}+r \mathbf{v})\right|{r=0}$,
> for deriving second-order derivatives in single fully connected layer, recurrent layer, and Stochastic
> Boltzmann Machine. However, the use of single-layer networks has become quite rare in contemporary
> times. Our focus is on the more complex Hessian submatrices across layers. Our method, OBA, facilitates
> the computation of the Hessian-vector product in multi-layer networks, which are the predominant form
> of neural networks today, marking a clear departure from the techniques used in "Fast Exact Multiplication
> by the Hessian".
>
> > **Question 1** Empirical evidence: While the results for CNNs are impressive, please consider transformers. There is much interest in pruning and redundancy in transformers e.g.
> a. Men, X., Xu, M., Zhang, Q., Wang, B., Lin, H., Lu, Y., Han, X. and Chen, W., 2024. Shortgpt: Layers in large language models are more redundant than you expect. arXiv preprint arXiv:2403.03853. b. Lad, V., Gurnee, W. and Tegmark, M., 2024. The Remarkable Robustness of LLMs: Stages of Inference?. arXiv preprint arXiv:2406.19384.
>
> Thank you for your recommendation! We have indeed performed experiments on ViT-B/16 and achieved
> promising results compared to magnitude pruning and first-order Taylor approximation (see Table 2).
> It would be worthwhile to conduct further experiments to test its effectiveness on other architectures
> such as BERT and GPT in future studies.
>
> > **Question 2** L652 Consider setting the scene, motivation and related work as an integral part of the paper. With due refence to early work on pruning and Hessian structure
>
> Thanks! We've revised the paper according to your suggestions. Please see the section 2 of our revised paper.
>
> > **Question 3** L058-9 The OBS method is only different from OBD when the non-diagonal Hessian is used, i.e they do not ignore the off-diagonal terms in OBS
>
> We apologize for the misunderstanding writing. OBS does not discard the second-order derivative information but rather
> approximates them with the Fisher information matrix. We've revised the content as follows:
> *They either discard or approximate the second-order partial derivatives between all pairs of parameters, which capture the change of loss on one parameter when deleting another parameter.*

---

> > ### Comment · Reviewer_fJfY · 2024-11-26
> > **Please argue why Pearlmutter's method is restricted to a single layer machine**
> >
> > Please argue why you think Pearlmutter's method (for computing the product of Hessian and vector) is restricted to a single layer machine. See also this ref:
> > Møller, M. (1993a). Exact calculation of the product of the Hessian matrix of feed-forward network error functions and
> > a vector in O(n) time. Daimi PB-432, Computer Science Department, Aarhus University, Denmark
> > for the same general result

---

> > > ### Author Response · Authors · 2024-11-26
> > > **Reply to Reviewer fJfY (Pearlmutter's method)**
> > >
> > > We are sorry that we misunderstood Pearlmutter's method for computing the product of Hessian and vector. After carefully reading the two papers, we agree that Pearlmutter's method can be applied to multiple layers including fully connected neural networks and recurrent neural networks.
> > >
> > > Our work extends the idea of exact Hessian-vector product computation to pruning, with a different Hessian-vector product calculation pipeline. This statement is added in the section 2 of our revised manuscript:
> > >
> > > "*Pearlmutter (1994) initially introduced an efficient method for computing the Hessian-vector product. Our research applies this idea to the pruning of modern network architectures including CNNs and Transformers.*"

---

> ### Author Response · Authors · 2024-11-20
> **Reply to Reviewer fJfY (2/2)**
>
> > **Question 4** L095 You mention that the Gauss.Newton approximation is in-sufficient what is the evidence?
>
> That's a good question. In our importance calculation process, the importance score of a weight $\theta_i$ is the sum of all second-order derivatives multiplied with corresponding parameters $\sum_{j}\frac{\partial^2 \mathcal{L}}{\partial\theta_i\partial\theta_j}\delta\theta_i\delta\theta_j$.
> If we take the approximation, $\sum_{j}\frac{\partial^2 \mathcal{L}}{\partial\theta_i\partial\theta_j}\delta\theta_i\delta\theta_j$ would turn into
> $\sum_{j}\frac{\partial \mathcal{L}}{\partial\theta_i}\frac{\partial \mathcal{L}}{\partial\theta_j}\delta\theta_i\delta\theta_j=\sum_{j}\frac{\partial \mathcal{L}}{\partial\theta_j}\delta\theta_j\frac{\partial \mathcal{L}}{\partial\theta_i}\delta\theta_i = c \frac{\partial \mathcal{L}}{\partial\theta_i}\delta\theta_i$, where c is a constant value. This importance score is same to that of Taylor method, which is outperformed by OBA in nearly all results.
>
> > **Question 5** L087 Do you keep track of the first order term (which is assumed zero in OBD/OBS)?
>
> We don't keep track of the first order term in our algorithm.

---

> > ### Comment · Reviewer_fJfY · 2024-11-26
> > **Significance of first order term?**
> >
> > In earlier work it was assumed that the network was trained to a minimum, i.e., with vanishing first term. You seem not to keep track of the first order term, hence potentially incur an uncontrolled error for networks trained with SGD methods. Please comment on this risk

---

> > > ### Author Response · Authors · 2024-11-26
> > > **Reply to Reviewer fJfY (first order term)**
> > >
> > > The reason for omitting the first-order term is to align with prior Hessian-based pruning methods. Previous studies assume that the network is trained to a minimum such that the first-order term vanishes. This assumption is indeed quite restrictive and nearly impossible to achieve in practical settings, where neural networks are typically trained to a local minimum using the SGD optimizer. However, we argue that while a local minimum does not reduce the first-order term to zero, it can still keep it relatively small.
> > >
> > > We performed experiments to record both first-order and second-order values for each layer in ResNet20 on CIFAR10 in unstructured pruning setting and compared the results. Our findings show that the average magnitude of the second-order term is roughly ten times that of the first-order term per layer, suggesting that the first-order term has little impact on our importance score calculation. Moreover, the unstructured pruning results do not differ significantly, indicating that the error introduced by omitting the first-order term is present but very limited.
> > >
> > > | Sparsity | Taylor (91%) |           | OBD (91%)    |           | Weight (91%) |            | OBA w/o First Order Term  (91%) |            | OBA w/. First Order Term (91%) |            |
> > > | :------: | :----------: | :-------: | :----------: | :-------: | :----------: | :--------: | :-----------------------------: | :--------: | :----------------------------- | :--------- |
> > > |          | Accuracy (%) | Ratio (%) | Accuracy (%) | Ratio (%) | Accuracy (%) | Ratio (%)  | Accuracy (%)                    | Ratio (%)  | Accuracy (%)                   | Ratio (%)  |
> > > | 0\.1     | 11\.00       | 14\.45    | 10\.03       | 13\.17    | 90\.66       | 99\.63     | 90\.83                          | **99\.81** | 90\.69                         | 99\.66     |
> > > | 0\.2     | 11\.00       | 14\.45    | 10\.03       | 13\.17    | 90\.82       | 99\.80     | 90\.90                          | **99\.89** | 90\.34                         | 99\.27     |
> > > | 0\.3     | 10\.00       | 13\.14    | 10\.03       | 13\.17    | 90\.67       | 99\.64     | 90\.65                          | 99\.62     | 90\.73                         | **99\.70** |
> > > | 0\.4     | 10\.80       | 14\.19    | 10\.02       | 13\.16    | 90\.67       | **99\.64** | 90\.35                          | 99\.29     | 90\.37                         | 99\.31     |
> > > | 0\.5     | 10\.00       | 13\.14    | 10\.01       | 13\.15    | 90\.79       | **99\.77** | 90\.57                          | 99\.53     | 90\.63                         | 99\.59     |
> > > | 0\.6     | 8\.20        | 10\.77    | 10\.00       | 13\.14    | 90\.23       | 99\.15     | 90\.69                          | **99\.66** | 90\.27                         | 99\.20     |
> > > | 0\.7     | 10\.00       | 13\.14    | 10\.00       | 13\.14    | 88\.83       | 97\.62     | 89\.94                          | 98\.84     | 89\.98                         | **98\.88** |
> > > | 0\.8     | 10\.00       | 13\.14    | 10\.00       | 13\.14    | 85\.03       | 93\.44     | 89\.64                          | **98\.51** | 88\.95                         | 97\.75     |
> > > | 0\.9     | 10\.00       | 13\.14    | 10\.52       | 13\.82    | 67\.00       | 73\.63     | 86\.27                          | 94\.80     | 86\.36                         | **94\.90** |

---

> > > > ### Comment · Reviewer_fJfY · 2024-12-02
> > > > **Why not simply include the first order term?**
> > > >
> > > > Thank you for the additional experiment to investigate the role of the first order term. Although you seem to have evidence that the term is small, I see no reason not to include it in the overall estimate (it is very limited additional compute, right?) for rigor and as a safety precaution, towards other applications where the term could be larger (e.g. when regularization is more important)

---

> > > > > ### Author Response · Authors · 2024-12-02
> > > > > **Discussion about adding the first order term**
> > > > >
> > > > > Thank you for your response! In our paper, we choose to omit the first-order term to ensure fair comparisons with other Hessian-based methods, which also do not incorporate it. However, we believe that in situations where the neural network is not fully trained, or when regularization plays a more important role, in order to make the pruning process theoretically safer and more rigorous, the first-order term can easily be reintroduced by adding a single line of code to our implementation. This discussion has been added to the revised paper, although it could not be uploaded due to the passing of the deadline.

---

> ### Author Response · Authors · 2024-12-02
> **Gentle Reminder**
>
> Dear reviewer fJfY,
>
> As the discussion period is nearing its end, we would like to know if you have any additional concerns regarding our paper. If so, we are happy to address them before the period concludes.
>
> Best,
>
> Authors

---

> ### Author Response · Authors · 2024-12-03
> **Our Summary on the First Order Term**
>
> Dear reviewer fJfY,
>
> we would like to summarize our opinions regarding the first-order term as follows.
>
> 1. **The reason for not incorporating the first order term is to ensure fair comparisons with other Hessian-based methods**, including OBD [1], OBS [2], EigenDamage [3], et al. They do not utilize the first-order term, either.
>
> 2. In our experiments, we found that the second-order term's average magnitude is approximately ten times greater than that of the first-order term per layer, indicating that the first-order term has minimal influence on the calculation of our importance score.
>
> 3. Through further experiments, as shown in the table presented in our "Reply to Reviewer fJfY (first order term)", we show that **including the first-order term does not have a clear influence on the performance of the pruned network**.
>
> 4. All our experiments do not leverage large regularization. Before pruning, all networks are fully trained into local minima, theoretically supporting the fact of the small first-order term.
>
> The first-order term can be added back in other conditions by minimal code adjustment. Under our experiment settings, not incorporating it is valid and fair. We hope our response could resolve your concerns regarding the first-order term.
>
> Best,
>
> Authors
>
> [1] LeCun, Yann, John Denker, and Sara Solla. "Optimal brain damage." Advances in neural information processing systems 2 (1989).
>
> [2] Hassibi, Babak, David G. Stork, and Gregory J. Wolff. "Optimal brain surgeon and general network pruning." IEEE international conference on neural networks. IEEE, 1993.
>
> [3] Wang, Chaoqi, et al. "Eigendamage: Structured pruning in the kronecker-factored eigenbasis." International conference on machine learning. PMLR, 2019.

---

### Official Review · Reviewer_1PJY · 2024-11-03

**Soundness:** 3
**Presentation:** 3
**Contribution:** 2
**Rating:** 6
**Confidence:** 2

**Summary:**

The authors propose a new method for neural network pruning called Optimal Brain Apoptosis (OBA) inspired by the prior work Optimal Brain Damage (Lecun et al., 1989). This method calculates the Hessian-vector product for each parameter and identify the conditions under which inter-layer Hessian submatrices are non-zero. The proposed method is able to prune models to increase model efficiency on 3 convolutional backbones and one vision transformers backbone on CIFAR10, CIFAR100 and ImageNet datasets.

**Strengths:**

1.	Pruning performance looks fairly promising. The method achieves consistent improvement on various datasets, using the most commonly-used backbones (ResNet and ViT), and on unstructured vs structured pruning.
2.	The results shown are quite comprehensive: pruning performance (accuracy, parameter reduction, FLOPs reduction, throughput increase), pruning cost (training and pruning time). Surprisingly, the pruning cost was not as high as I originally expected knowing that the method involves computation of the Hessian-vector product.

**Weaknesses:**

1.	Since pruning is not my area of expertise, I am unsure whether the authors used fair baselines for comparison. The proposed method is mainly compared against 7 methods from 3 papers, respectively in 2016, 2017 and 2019. A simple literature search gave me a few methods that claim to have achieved better pruning performances and are fairly well cited and fairly highly stared: https://arxiv.org/abs/2203.04248, https://arxiv.org/abs/2208.11580, https://arxiv.org/abs/2210.04092, https://arxiv.org/abs/2112.00029. I would encourage the authors to either compare to some of the latest works, or provide a justification on not considering these more recent works --- meanwhile, I would resort to reviewers with more experience in pruning on their opinions regarding this matter.

**Questions:**

1.	Please refer to Weakness 1.
2.	Minor suggestion. For LaTeX notations, the subscripts (especially subscripts of superscripts) can be wrapped in text format. What I mean is instead of $\mathbb{R}^{l_out}$, using $\mathbb{R}^{l_\textrm{out}}$ might give you a better looking symbol.
3.	What do a, b, c, d, e respectively mean in Equation 3? Could the authors define them or point to the text where they are defined?

---

> ### Author Response · Authors · 2024-11-20
> **Reply to Reviewer 1PJY (1/2)**
>
> We express our gratitude for your recognition on our work.
> > **Weakness & Question 1** Since pruning is not my area of expertise, I am unsure whether the authors used fair baselines for comparison. The proposed method is mainly compared against 7 methods from 3 papers, respectively in 2016, 2017 and 2019. A simple literature search gave me a few methods that claim to have achieved better pruning performances and are fairly well cited and fairly highly stared: https://arxiv.org/abs/2203.04248, https://arxiv.org/abs/2208.11580, https://arxiv.org/abs/2210.04092, https://arxiv.org/abs/2112.00029. I would encourage the authors to either compare to some of the latest works, or provide a justification on not considering these more recent works --- meanwhile, I would resort to reviewers with more experience in pruning on their opinions regarding this matter.
>
> Thank you for providing additional related work in the field of pruning. Our paper primarily introduces a novel
> Hessian-vector product method, where the Hessian matrix represents the second-order term in the Taylor
> expansion of the loss function. As such, we mainly focus our comparisons on works that similarly utilize
> either the Hessian matrix or the first-order term of the Taylor expansion. The seven methods you highlighted
>  are from Table 3, which presents structured pruning results. For unstructured pruning, our results are also
>   compelling, as shown in Tables 4 and 5, particularly against CHITA [1].
>
> In the paper you recommended, both "Dual Lottery Ticket Hypothesis" and "Advancing Model Pruning via
> Bi-level Optimization" build on the Lottery Ticket Hypothesis, which seeks to identify an effective
> subnetwork prior to training, a concept distinct from our approach. "Pixelated Butterfly" does not use
> Hessian or Taylor expansion information, thus it is not a relevant comparison for our work. "Optimal
> Brain Compression," which utilizes the Hessian matrix for pruning, is indeed worth comparing. However,
> we were unable to replicate the author's results in our implementation (Our GMP achieves 73.16% accuracy
> under the same conditions described in OBC, which is significantly lower than the reported 74.86%), and
> the time constraints of the rebuttal period prevent us from implementing OBA within the OBC structure.
> We have expanded our unstructured pruning results on CIFAR10 using Resnet20 and included CBS [2] and WoodFisher [3] in our
> comparisons.
>
>
> | Weight (91%) |            | WoodFisher (91\.36%) |             | CBS (91\.36%) |           | Chita++ (91\.36%) |            | OBA (91%)   |            |
> | :----------- | :--------- | :--------------------- | :---------- | :-------------- | :-------- | :------------------ | :--------- | :----------- | :--------- |
> | Accuracy (%) | Ratio (%)  | Accuracy (%)           | Ratio (%)   | Accuracy (%)    | Ratio (%) | Accuracy (%)        | Ratio (%)  | Accuracy (%) | Ratio (%)  |
> | 90\.66       | 99\.63     | -                      | -           | -               | -         | -                   | -          | 90\.83       | **99\.81** |
> | 90\.82       | 99\.80     | -                      | -           | -               | -         | -                   | -          | 90\.90       | **99\.89** |
> | 90\.67       | 99\.64     | 91\.37                 | **100\.01** | 91\.35          | 99\.99    | 91\.25              | 99\.88     | 90\.65       | 99\.62     |
> | 90\.67       | 99\.64     | 91\.15                 | 99\.77      | 91\.21          | 99\.84    | 91\.20              | **99\.82** | 90\.35       | 99\.29     |
> | 90\.79       | **99\.77** | 90\.23                 | 98\.76      | 90\.58          | 99\.15    | 91\.04              | 99\.65     | 90\.57       | 99\.53     |
> | 90\.23       | 99\.15     | 87\.96                 | 96\.28      | 88\.88          | 97\.29    | 90\.78              | 99\.37     | 90\.69       | **99\.66** |
> | 88\.83       | 97\.62     | 81\.05                 | 88\.71      | 81\.84          | 89\.58    | 90\.38              | **98\.93** | 89\.94       | 98\.84     |
> | 85\.03       | 93\.44     | 62\.63                 | 68\.55      | 51\.28          | 56\.13    | 88\.72              | 97\.11     | 89\.64       | **98\.51** |
> | 67\.00       | 73\.63     | 11\.49                 | 12\.58      | 13\.68          | 14\.97    | 79\.32              | 86\.82     | 86\.27       | **94\.80** |
>
> As demonstrated, OBA significantly outperforms other methods at high sparsity levels, confirming its efficacy.
>
> [1] Benbaki, Riade, et al. "Fast as chita: Neural network pruning with combinatorial optimization." In ICML 2023.
>
> [2] Yu, Xin, et al. "The combinatorial brain surgeon: pruning weights that cancel one another in neural networks." In ICML 2022.
>
> [3] Singh, Sidak Pal, and Dan Alistarh. "Woodfisher: Efficient second-order approximation for neural network compression." In NeurIPS 2020.

---

> ### Author Response · Authors · 2024-11-20
> **Reply to Reviewer 1PJY (2/2)**
>
> > **Question 2** Minor suggestion. For LaTeX notations, the subscripts (especially subscripts of superscripts)
> can be wrapped in text format. What I mean is instead of $\mathbb R^{l_{out}}$, using $\mathbb R^{l_{\text{out}}}$
>  might give you a better looking symbol.
>
>  Thanks for your helpful suggestions. Wrapping these notations with "\text" do present a better looking. We've changed
>  all these symbols into "\text{}" version in the revised manuscript.
>
>  > **Question 3** What do a, b, c, d, e respectively mean in Equation 3? Could the authors define them or point to the text where they are defined?
>
> a, b, c, d, e represent the index of 5 dimensions, respectively the output channel/neuron dimension with length $l_{\text{out}}$,
> the flattened output feature size dimension with length $p_{\text{out}}$, the input channel/neuron dimension with length $l_{\text{in}}$,
> the flattened weight size for each input and output neuron/channel pair with length $p_{\text{weight}}$, and the
> flattened input feature size dimension with length $p_{\text{input}}$. You could refer to the beginning part of section 3.1 to better understand them with concrete examples.

---

> ### Comment · Reviewer_1PJY · 2024-11-24
> **Response to rebuttal**
>
> The authors provided a disciplined response to my prior question on why they did not include comparisons against certain more recent pruning methods, and I am satisfied with the answer. Under the current form of the paper, I do not see pressing reasons for a major rating increase from 6 marginal accept to 8 clear accept, and I would feel more comfortable keeping my original rating. Nevertheless, I acknowledge the extra effort spent by the authors and wish them all the best.

---

> > ### Author Response · Authors · 2024-11-25
> > **Reply to Reviewer 1PJY's comment**
> >
> > Dear reviewer 1PJY,
> >
> > We appreciate your final recognition of our efforts in the revised manuscript and rebuttal. Hope all is well with you!
> >
> > Best,
> >
> > Authors

---

### Official Review · Reviewer_v9aH · 2024-11-04

**Soundness:** 3
**Presentation:** 3
**Contribution:** 3
**Rating:** 6
**Confidence:** 3

**Summary:**

This paper introduces a tractable way of computing the Hessian of the loss function in feed-forward networks. This finding is very interesting, as it may have a wide range of applications, in particular, the one of pruning CNNs and transformers. The results are very promising, as they show some improvements, but these improvements do not seem to be very large.

**Strengths:**

An interesting new method for computing the Hessian of the loss function in feed-forward networks in a tractable way.

**Weaknesses:**

The improvements in accuracy and speed do not seem overwhelming.

**Questions:**

Considering that recurrent neural networks use back-propagation through time, for a given time window, how tractable would be to extend your method to such networks?

---

> ### Author Response · Authors · 2024-11-20
> **Reply to Reviewer v9aH (1/1)**
>
> Thanks for your reognition on our work!
> > **Weakness** The improvements in accuracy and speed do not seem overwhelming.
>
> We acknowledge your concerns regarding some of the results presented. Specifically, in Table 2, you will notice that our method, OBA, shows significantly better results for structured pruning on the ViT-B/16 model compared to other methods. In terms of unstructured pruning at high sparsity levels, as shown in Tables 4 and 5, OBA notably outperforms CHITA and other competing approaches. However, we realize that the superiority of our method is less pronounced in Figure 3 and Table 3, though it still maintains a marginal advantage over other methods. Thank you for bringing this to our attention.
>
> > **Question** Considering that recurrent neural networks use back-propagation through time, for a given time window, how tractable would be to extend your method to such networks?
>
> In an RNN, the concept of time is incorporated, requiring an expansion of series and parallel connectivity in networks.
> For layers that do not convey temporal information, series and parallel connectivity are confined to individual time steps.
> For recurrent neurons that propagate information through time, such as the basic RNN neurons whose hidden state is updated as
> $$h_t=\tanh \left(W_{h h} h_{t-1}+W_{x h} x_t+b_h\right)$$
> and
> $$y_t=W_{h y} h_t+b_y,$$
> the output $y_{t'}$ at time $t'$ is dependent on $x_{\tau}$ where $\tau \leq t'$. This dependency causes layers from earlier time steps, up to $t'$, to be in series connectivity with all subsequent layers at time step $t'$.
> Furthermore, since the parameters $W_{hh}$, $W_{xh}$, $W_{yh}$ interact to generate $y_t$,
> their Hessian submatrix between them is non-zero and must be calculated. The Hessian computation for a State Space Model is analogous to that in a basic RNN.
>
> For more intricate recurrent layers like LSTMs, which incorporate multiple gates that affect the internal state,
> the network displays extensive parallel connectivities.
> Computing the Hessian for such a structure is significantly more complex.
> Nonetheless, through a systematic and categorized approach,
> we can still effectively address the complexities introduced by these advanced neurons using OBA. These are directions worth exploring in the future.

---

> > ### Comment · Reviewer_v9aH · 2024-11-25
> > **Thank you for your answer**
> >
> > I will keep my score

---

> > > ### Author Response · Authors · 2024-11-25
> > > **Appreciation from Authors**
> > >
> > > Dear Reviewer v9aH,
> > >
> > > Thanks for reviewing our response. Wish you all the best!
> > >
> > > Authors

---

### Official Review · Reviewer_rbue · 2024-11-07

**Soundness:** 4
**Presentation:** 4
**Contribution:** 4
**Rating:** 8
**Confidence:** 4

**Summary:**

This paper presents Optimal Brain Apoptosis (OBA), which appears to be a novel pruning method for neural networks that builds upon the principles of Optimal Brain Damage (OBD), which date back to 1980s and 1990s. Recognizing the limitations of previous pruning methods, OBA leverages the full Hessian-vector product to compute the importance of each parameter in the network accurately, moving beyond previous methods that relied on approximations. The authors first analyze the conditions under which Hessian submatrices between layers are nonzero and develop an approach to calculate the second-order Taylor expansion for each parameter (enabling precise pruning in both structured and unstructured settings). Empirical tests demonstrate that OBA effectively reduces computational overhead while maintaining model accuracy. The authors acknowledge that while OBA works well for architectures like CNNs and Transformers, extending it to more complex models like RNNs or State Space Models will require further research.

**Strengths:**

- Unlike previous methods that approximate the Hessian matrix, OBA calculates the full Hessian-vector product, providing a more accurate measure of parameter importance and leading to more precise pruning.
- OBA supports both structured pruning (removing entire neurons, channels, or layers) and unstructured pruning (removing individual weights). It's compatible with a wide range of architectures.
- The approach optimizes the calculation of the Hessian-vector product (reduced computational complexity).
- Showing adaptability to widely-used architectures in deep learning.
- The authors provide a solid theoretical basis for their approach, with clear proofs.

**Weaknesses:**

- It seems like calculating the full Hessian-vector product, even with optimizations, can still be computationally expensive for larger and more complex networks.
- Extending this method to newer architectures seems to require additional work.
-The method was tested on a specific set of architectures, and its generalizability across a wider range of tasks or domains is yet to be fully established.
- The experiments make the paper feel somewhat outdated, as the evaluations and datasets used are reminiscent of those from 7-8 years ago.

**Questions:**

Please see the weaknesses section. Providing feedback on those comments during rebuttal will be appreciated.

---

> ### Author Response · Authors · 2024-11-20
> **Reply to Reviewer rbue(1/1)**
>
> We deeply appreciate your recognition on our work! We would like to reply your questions and weaknesses as follows.
> > **Weakness 1** It seems like calculating the full Hessian-vector product, even with optimizations, can still be computationally expensive for larger and more complex networks.
>
> You are right. We recognize that our method for calculating the Hessian-vector product is still quite time-consuming, taking twice as long as OBS, as shown in Figure 6b. However, the time it takes for OBA to prune is still relatively minor compared to the extensive time required for training or fine-tuning the model. For very large models such as LLaMA 3.1, BLOOM, and others, it may be practical to prune only specific layers, similarly to common fine-tuning practices. This approach significantly reduces complexity by limiting the calculation of the Hessian-vector product to just the selected layers.
>
> > **Weakness 2** Extending this method to newer architectures seems to require additional work. The method was tested on a specific set of architectures, and its generalizability across a wider range of tasks or domains is yet to be fully established.
>
> Thanks for pointing out this. OBA can be applied to various parameter layers such as fully connected, convolutional, and attention layers, which are essential to most contemporary models in computer vision and natural language processing. This versatility shows that OBA can be seamlessly integrated into widely used models without any modifications. Moreover, since the importance score acquisition of OBA only relies on the loss of the output, which is task-invariant, OBA can also be directly utilized in other tasks to prune models. Exploring the application of the Hessian-vector product in other architectures like RNNs, SSMs, and additional tasks will be valuable to further assess OBA’s adaptability and generalization capabilities in future research.
>
> > **Weakness 3** The experiments make the paper feel somewhat outdated, as the evaluations and datasets used are reminiscent of those from 7-8 years ago.
>
> We appreciate your feedback and sincerely apologize if our experiments gave the impression of being outdated. Our intention was to compare our method fairly with others that also innovate on the Taylor expansion term, specifically Eigen Damage (2019) and CHITA (2023). Eigen Damage is a structured pruning method, while CHITA focuses on unstructured pruning. Although CHITA is more recent, the models and datasets used in both studies are quite similar, which may have contributed to the perception of our experiments being less current.
>
> For fairness, we conducted our experiments under the same settings as these studies, ensuring a direct and meaningful comparison. However, we understand how this might make the experiments appear outdated. Moving forward, we plan to explore the application of OBA on more contemporary datasets and models to demonstrate its broader applicability and relevance. Thank you for pointing this out, and we will address it in our future work.

---

> > ### Comment · Reviewer_rbue · 2024-11-26
> >
> > Thank you for your responses! I will keep my score.

---

> > > ### Author Response · Authors · 2024-11-26
> > > **Appreciation for Reviewing our Rebuttal**
> > >
> > > Dear Reviewer rbue,
> > >
> > > Thanks for taking the time to review our response and recognizing our work! Wish you a great day!
> > >
> > > Best,
> > >
> > > Authors

---

### Author Response · Authors · 2024-11-20
**Reply to All Reviewers**

Thank you for your thorough reviews and valuable feedback. We are grateful for the opportunity to address the concerns raised and clarify aspects of our work.

We've carefully revised our manuscript according to the reviewer's valuable suggestions. The new content is highlighted in orange.

---

### Meta-Review · Area_Chair_9VuC · 2024-12-17

**Metareview:**

The paper proposes a pruning method inspired by work on Optimal Brain Damage. The method estimates the importance of parameters based on the Hessian matrix, and the method computes the Hessian-vector product directly for each parameter, providing a more accurate prediction or parameter importance than using approximations that most prior methods use.

* Reviewer rbue notes that the method is applicable to widely used architectures and theoretically grounded with proofs. However, also notes that the method is computationally expensive and the experiments use outdated architectures and datasets. The authors acknowledge that the method is computationally expensive and note that their experimental setup is to facilitate comparisons with the literature.

* The review by v9aH lacked sufficient detail and could not be fully considered in the meta review.

* Reviewer 1PJY notes that performance looks promising and the results are comprehensive. The reviewer notes that the comparisons are with works more than 5 years old, even though there are a variety of recent ones. The authors note that comparing to Hessian-based related work is sufficient; I disagree with the authors here, the paper would benefit from a comparison to non-Hessian based pruning methods, or at least a discussion on how the method compares to those.

* fJfY finds the related work not sufficient. The authors revised the related literature.

The proposed method is interesting, however based on my own reading the theory is relatively shallow, and based on the reviewer's feedback the paper would benefit from a more solid experimental setup, in particular comparisons to state-of-the-art methods on current neural network architectures.

**Additional Comments On Reviewer Discussion:**

see meta review

---

### Decision · Program_Chairs · 2025-01-22

Accept (Poster)